# DESIGN: ENCRYPTED GNN INFERENCE VIA SERVER-SIDE INPUT GRAPH PRUNING

## ABSTRACT

Graph Neural Networks (GNNs) enable powerful graph learning, yet Fully Homomorphic Encryption (FHE) makes inference prohibitively expensive. We present DESIGN (EncrypteD GNN Inference via sErver-Side Input Graph pruNing), a server-side framework that reduces FHE cost without client changes. DESIGN computes encrypted degree–based importance scores and uses homomorphic comparisons to produce multi-level masks, which drive two optimizations: logical pruning of low-importance nodes and edges, and importance-aware assignment of low-degree polynomial activations to most nodes while reserving higher degrees for critical ones. Across standard benchmarks, DESIGN substantially accelerates encrypted GNN inference while maintaining competitive accuracy. Code is available at `https://anonymous.4open.science/r/DESIGN-7F93`.

## 1 INTRODUCTION

Graph Neural Networks (GNNs) have become increasingly popular tools for learning from graph-structured data, demonstrating remarkable performance in diverse real-world applications such as recommendation systems, bioinformatics, and social network analysis (Gao & Ji, 2019; Chen et al., 2025; Zheng et al., 2020; Wu et al., 2020; 2022). The success of GNNs primarily stems from their inherent ability to capture complex relational information and structural patterns through message-passing mechanisms between nodes (Gilmer et al., 2017). However, despite their widespread adoption, the use of GNNs on sensitive graph data, which often contains personal or proprietary information (e.g., user profiles, financial transactions, social connections), raises significant privacy concerns, especially when processed by third-party cloud services (Zhang et al., 2024b; Ju et al., 2024; Zhao et al., 2025a; Cao et al., 2011; Zhao et al., 2025b; Cheng et al., 2025; Wang et al., 2025). Such processing can expose confidential details to potential breaches and unauthorized access (Wang et al., 2023). Fully Homomorphic Encryption (FHE) perfectly aligns with the need for privacy in this domain, as it allows computations to be performed directly on encrypted data (ciphertexts) without requiring decryption by the server, thus ensuring end-to-end data confidentiality (Gentry, 2009; Brakerski et al., 2014). Consequently, applying FHE to GNNs has emerged as a critical research direction to enable secure and private graph analytics (Ran et al., 2022; Peng et al., 2023; Effendi & Chattopadhyay, 2024; Huang et al., 2024). Nevertheless, this combination introduces substantial computational challenges. Basic arithmetic operations under FHE are orders of magnitude more expensive than their unencrypted, i.e., plaintext, counterparts (Acar et al., 2018; Asharov et al., 2012; Cheon et al., 2017). This overhead, particularly from numerous homomorphic multiplications and complex polynomial approximations for non-linearities, makes direct FHE translation of GNN inference prohibitively slow for many practical applications, creating an urgent need for techniques that enhance the efficiency while preserving both privacy and utility.

Researchers have explored several approaches to address this privacy-efficiency challenge. Recent methods have primarily focused on optimizing the GNN model's internal structure, reducing homomorphic operations through adjustments to model architectures and polynomial approximations of activation functions (Ran et al., 2022). These approaches exploit sparsity within adjacency matrices and simplify activation functions to significantly lower computational complexity. Other techniques include statistical pruning methods (Yik et al., 2022; Liu et al., 2023; Liao et al., 2024; Chen et al., 2025) that have shown promise in plaintext environments but remain underexplored in encrypted domains. Model-centric optimizations typically overlook the inherent redundancy present within the input graph data itself (Jia et al., 2020; Tan et al., 2019). Moreover, existing approaches often apply

uniform optimization strategies across all graph components (Panda, 2022), missing opportunities for adaptive resource allocation based on node and edge importance. The exploration of efficient privacy-preserving GNN inference techniques that leverage both model and data characteristics thus remains nascent, with substantial gaps in balancing privacy, efficiency, and accuracy.

Nevertheless, despite the urgent need for efficient privacy-preserving GNN inference, achieving this under server-side FHE constraints is a non-trivial task, and we identify three fundamental challenges. Operating within the stringent constraints of server-side FHE inference, where users upload fully encrypted graphs and the server cannot decrypt data, presents unique challenges for achieving efficiency through pruning and adaptive computation. We identify three primary challenges: (1) *Impracticality of Online Encrypted Pruning.* Performing dynamic, data-dependent pruning directly on ciphertexts is largely infeasible. The server operates without access to plaintext graph metrics (e.g., node degrees, feature norms). Attempting to compute even simple proxies for these metrics homomorphically incurs significant FHE overhead, particularly from multiplications, and subsequent comparisons against thresholds require expensive HE comparison protocols. Consequently, most existing graph pruning techniques (Yik et al., 2022; Liu et al., 2023; Liao et al., 2024; Chen et al., 2025; Gao & Ji, 2019; Zheng et al., 2020; Yu et al., 2021; Chen et al., 2021), reliant on plaintext access or efficient comparisons, are unsuitable for online, server-side FHE application. (2) *Inefficient Uniform Activation Approximation.* Efficiently handling non-linear activation functions under FHE remains a critical bottleneck. While polynomial approximations are necessary (Ran et al., 2022), existing methods typically apply a uniform polynomial degree across all nodes. This is suboptimal, as a high degree for critical nodes dictates overall cost, while a uniform low degree can degrade accuracy. The challenge is to enable adaptive polynomial allocation based on node importance without costly online FHE comparisons. (3) *Ensuring GNN Architecture Generality.* Developing an efficiency framework applicable across diverse GNN architectures is challenging. An ideal solution should enhance performance without tight coupling to specific GNN types or requiring model modifications, ensuring wider usability.

To address the challenges above, in this paper, we propose a principled framework named DESIGN (EncrypteD GNN Inference via sErver-Side Input Graph pruNing). Specifically, DESIGN offers key advantages by enabling server-side efficiency enhancements for FHE GNN inference without requiring client-side modifications or compromising the core privacy guarantees of FHE. It achieves this by intelligently managing computational resources based on dynamically assessed data importance directly under FHE. To handle the challenge of the *Impracticality of Online Encrypted Pruning*, DESIGN computes FHE-compatible importance scores (e.g., based on encrypted node degrees inspired by methods like (Yik et al., 2022; Liu et al., 2023; Liao et al., 2024; Chen et al., 2025)) from the received encrypted graph. These encrypted scores are then used with approximate homomorphic comparison protocols to generate multi-level importance masks directly on the server. This allows for logical pruning of unimportant elements and significantly decreases the volume of data requiring homomorphic computation, thereby addressing FHE overhead, albeit with the computational cost associated with homomorphic comparisons. To tackle the challenge of *Inefficient Uniform Activation Approximation*, DESIGN introduces an importance-aware activation function allocation strategy. Guided by the dynamically generated importance levels (encoded in the masks), polynomial approximations of varying degrees are assigned: high-degree polynomials (extending concepts from (Ran et al., 2022)) are used for critical nodes to preserve accuracy, while lower-degree approximations are applied to less important nodes, reducing ciphertext multiplications and overall computational cost. Finally, to address the challenge of *Ensuring GNN Architecture Generality*, DESIGN's core mechanisms, dynamically generating importance masks from encrypted data and selecting polynomial degrees based on these mask levels are independent of specific GNN layer operations. This independence allows the framework to be integrated with diverse GNN architectures without requiring significant model-specific alterations.

In summary, the contributions of this paper are three-fold:

- **Novel Dual-Pruning Framework:** We propose DESIGN, a new framework introducing a *dual-pruning scheme* for encrypted GNN inference. This scheme combines (1) input graph data pruning, by removing less important nodes and edges, with (2) GNN model-level adaptation, through adaptive polynomial activations.

- **Adaptive Polynomial Activations:** DESIGN dynamically computes encrypted node importance scores under FHE, which then guide the server-side generation of level masks for selecting varied-degree polynomial activations, optimizing the cost-accuracy balance.

- **Experimental Evaluation:** We will conduct extensive experiments to demonstrate the effectiveness of DESIGN, quantifying its inference speedups and accuracy impact against foundational and state-of-the-art FHE GNN frameworks on benchmark datasets.

## 2 PRELIMINARIES

**Notations.** We use bold uppercase letters (e.g., $\mathbf{A}$) for matrices, bold lowercase letters (e.g., $\mathbf{x}$) for vectors, and normal lowercase letters (e.g., $n$) for scalars. An attributed graph is $\mathcal{G} = (\mathcal{V}, \mathcal{E}, \mathbf{X})$, where $\mathcal{V} = \{v_1, \ldots, v_n\}$ is the set of $n$ nodes, $\mathcal{E} \subseteq \mathcal{V} \times \mathcal{V}$ is the set of edges, and $\mathbf{X} \in \mathbb{R}^{n \times d_0}$ is the initial node feature matrix with feature dimensionality $d_0$. The adjacency matrix is $\mathbf{A} \in \{0, 1\}^{n \times n}$ (or $\mathbf{A} \in \mathbb{R}^{n \times n}$ for weighted graphs), where $A_{uv} > 0$ indicates an edge from node $u$ to node $v$; the degree of node $v$ is $\deg(v)$. A Graph Neural Network model is $f_{\boldsymbol{\theta}}$, parameterized by $\boldsymbol{\theta}$, with $L$ layers. Node representations at layer $l$ are $\mathbf{H}^{(l)} \in \mathbb{R}^{n \times d_l}$, where $d_l$ is the hidden dimension and $\mathbf{H}^{(0)} = \mathbf{X}$; the pre-activation output of layer $l$ is $\mathbf{Z}^{(l+1)}$. A non-linear activation function is $\sigma(\cdot)$, and its degree-$k$ polynomial approximation is $P_k(\cdot)$. For Fully Homomorphic Encryption (FHE), $\text{Enc}(\cdot)$ and $\text{Dec}(\cdot)$ denote encryption and decryption. Encrypted data is marked with a tilde (e.g., $\tilde{\mathbf{X}} = \text{Enc}(\mathbf{X})$, $\tilde{\mathbf{A}} = \text{Enc}(\mathbf{A})$). Homomorphic addition, multiplication, and element-wise multiplication are $\oplus$, $\otimes$, and $\odot$, respectively. We denote specific homomorphic operations as follows: `HE.Add` for homomorphic addition, `HE.Mult` for homomorphic ciphertext-ciphertext or ciphertext-plaintext multiplication, `HE.Rotate` for homomorphic rotation (typically for SIMD operations), `HE.AprxCmp` for approximate homomorphic comparison (e.g., evaluating a polynomial approximation of a comparison function), and `HE.PolyEval` for the homomorphic evaluation of a pre-defined polynomial on an encrypted input. An encrypted vector of ones is $\tilde{\mathbf{1}}$. Importance scores for nodes are denoted by the vector $\mathbf{s}$ (plaintext) or $\tilde{\mathbf{s}}$ (encrypted). Importance thresholds are $\tau_1, \ldots, \tau_m$ (plaintext) or $\tilde{\tau}_1, \ldots, \tilde{\tau}_m$ (encrypted). Encrypted binary masks for pruning and importance levels are $\tilde{M}_0, \tilde{M}_1, \ldots, \tilde{M}_m$, respectively.

**Fully Homomorphic Encryption.** Fully Homomorphic Encryption (FHE) is a cryptographic technique enabling computation directly on encrypted data, thereby preserving data confidentiality during processing by untrusted servers. This capability is particularly valuable for privacy-preserving machine learning. Among various FHE schemes, the Cheon-Kim-Kim-Song (CKKS) scheme (Cheon et al., 2017) is widely adopted for such applications due to its native support for approximate arithmetic on encrypted real numbers. In the CKKS scheme, given plaintexts (i.e., unencrypted data) $a$ and $b$, their respective ciphertexts are denoted as $\tilde{a} = \text{Enc}(a)$ and $\tilde{b} = \text{Enc}(b)$. The scheme supports homomorphic addition ($\oplus$) and multiplication ($\otimes$) such that upon decryption $\text{Dec}(\cdot)$, $\text{Dec}(\tilde{a} \oplus \tilde{b}) \approx a + b$ and $\text{Dec}(\tilde{a} \otimes \tilde{b}) \approx a \cdot b$. A critical characteristic of FHE is that homomorphic multiplication ($\otimes$) significantly increases the noise level inherent in ciphertexts and is computationally more intensive than homomorphic addition. The cumulative noise and the maximum number of sequential multiplications (multiplicative depth) directly influence the choice of encryption parameters and overall computational complexity.

The application of Fully Homomorphic Encryption (FHE) to Graph Neural Network (GNN) inference, while ensuring data privacy, is significantly hampered by the inherent computational overhead of homomorphic operations. Our work addresses this within a server-side inference setting: a client encrypts their graph $\mathcal{G} = (\mathcal{V}, \mathcal{E}, \mathbf{X})$ into $\tilde{\mathcal{G}} = (\text{Enc}(\mathcal{V}), \text{Enc}(\mathcal{E}), \text{Enc}(\mathbf{X}))$ and uploads it; a server with a pre-trained GNN model $f_{\boldsymbol{\theta}}$ computes the encrypted result $\tilde{\mathbf{Y}} = f_{\boldsymbol{\theta}}(\tilde{\mathcal{G}})$ entirely homomorphically using operations like homomorphic addition $\oplus$ and multiplication $\otimes$. This encrypted result is then returned to the client for decryption $\text{Dec}(\cdot)$. A primary unaddressed challenge in this paradigm is efficiently mitigating the impact of input graph redundancy, as the server cannot directly analyze the encrypted $\tilde{\mathcal{G}}$ using conventional methods without incurring prohibitive FHE costs. This motivates the need for novel strategies to adapt the computation based on input characteristics while preserving privacy and efficiency, leading to the core research challenge formally defined in Problem 1.

**Problem 1.** *Efficient GNN Inference Under FHE. Given a pre-trained GNN model $f_{\boldsymbol{\theta}}$, an encrypted input graph $\tilde{\mathcal{G}} = Enc_{pk}(\mathcal{G})$ (where $\mathcal{G} = (\mathcal{V}, \mathcal{E}, \mathbf{X})$ is the plaintext graph and pk is the FHE public*

*key), an FHE scheme defined by the tuple (KeyGen, Enc, Dec, Eval, $\oplus, \otimes$) along with evaluation keys evk, and a user-specified maximum acceptable accuracy degradation $\epsilon > 0$, our goal is to design an FHE evaluation strategy $\hat{f}_{\boldsymbol{\theta}}$ such that the homomorphic evaluation $\tilde{\mathbf{Y}}' = \hat{f}_{\boldsymbol{\theta}}(\tilde{\mathcal{G}}, evk)$ achieves significantly minimized computational latency $Time(\hat{f}_{\boldsymbol{\theta}}(\tilde{\mathcal{G}}, evk))$ compared to a direct homomorphic evaluation $Time(Eval(f_{\boldsymbol{\theta}}, \tilde{\mathcal{G}}, evk))$, while ensuring that the decrypted result $\mathbf{Y}' = Dec_{sk}(\tilde{\mathbf{Y}}')$ (where sk is the FHE secret key) maintains comparable utility to the plaintext inference result $\mathbf{Y} = f_{\boldsymbol{\theta}}(\mathcal{G})$, i.e., $Accuracy(\mathbf{Y}') \geq Accuracy(\mathbf{Y}) - \epsilon$. All server-side computations must uphold the privacy guarantees inherent to the FHE scheme.*

## 3 METHODOLOGY

This section introduces our proposed framework, DESIGN, for efficient GNN inference under FHE, designed for server-side operation on client-encrypted data. DESIGN enhances efficiency through two primary stages: first, an *FHE-Compatible Statistical Importance Scoring and Partitioning* stage (detailed in Section 3.1), and second, an *FHE GNN Inference with Pruning and Adaptive Activation Allocation* stage (detailed in Section 3.2). The intuition for the first stage is to assess node importance directly on the encrypted graph using FHE-friendly statistics, such as node degree. This allows the server to differentiate node criticality without decryption, and its advantage lies in enabling subsequent targeted optimizations by identifying less crucial graph components, despite the inherent cost of homomorphic comparisons for partitioning. The intuition for the second stage is to leverage these pre-determined importance levels to reduce computational load during GNN inference. This brings the advantage of both logically pruning the least important elements (reducing data volume) and adaptively assigning computationally cheaper, lower-degree polynomial activation functions to less critical nodes while reserving higher-degree approximations for important ones (optimizing activation complexity), thereby accelerating inference while aiming to preserve accuracy.

### 3.1 FHE-COMPATIBLE STATISTICAL IMPORTANCE SCORING AND PARTITIONING

The initial stage of our framework, detailed in Appendix C.1 (Algorithm 1), assesses node importance and partitions the nodes of the input graph into distinct levels. This process operates directly on the encrypted graph $\tilde{\mathcal{G}} = (\text{Enc}(\mathcal{V}), \text{Enc}(\mathcal{E}), \tilde{\mathbf{X}})$ under the CKKS FHE scheme and involves two main steps: first, an *Encrypted Node Degree Computation* to derive an FHE-compatible importance score for each node, and second, a *Homomorphic Partitioning and Mask Generation* step to categorize nodes based on these scores. Given the encrypted adjacency matrix $\tilde{\mathbf{A}}$, we first compute an encrypted importance score vector $\tilde{\mathbf{s}}$. The selection of an appropriate importance metric is crucial, balancing informativeness with computational feasibility under FHE. While metrics based on node features (e.g., L2 norm, variance) could potentially offer richer importance signals, their computation under FHE, as illustrated in Appendix C.6 (Table 16), necessitates homomorphic multiplications (`HE.Mult`), leading to significantly higher computational costs, increased multiplicative depth, and greater noise accumulation. Such an overhead for the initial scoring could negate the benefits of subsequent pruning. Therefore, to prioritize FHE efficiency and minimize the complexity of this initial stage, our framework adopts node degree as the primary statistical indicator for importance, as its computation relies predominantly on homomorphic additions (`HE.Add`), which are far more efficient under FHE. This design choice allows for a lightweight initial assessment of node importance directly on the encrypted data, forming a practical basis for subsequent partitioning and adaptive inference.

**Encrypted Node Degree Computation.** To ensure FHE feasibility and reduce computational overhead, we utilize node degree as the primary statistical indicator for importance. Although node degree is inherently an integer, it is encoded as an approximate real number within the CKKS scheme for compatibility with subsequent approximate arithmetic operations. Assuming $\tilde{A}_{uv}$ (an element of the encrypted adjacency matrix $\tilde{\mathbf{A}}$) encrypts a value approximating 1 if an edge exists from node $u$ to node $v$, and 0 otherwise, the encrypted degree $\tilde{s}_v$ for node $v$ (an element of the encrypted score vector $\tilde{\mathbf{s}}$) is computed via homomorphic summation as $\tilde{s}_v = \bigoplus \tilde{A}_{vu}$, where $u \in \mathcal{V}$. This operation predominantly involves efficient homomorphic addition (`HE.Add`) and potentially homomorphic rotation (`HE.Rotate`) if SIMD (Single Instruction, Multiple Data) packing is employed to batch multiple adjacency matrix entries or partial sums within ciphertexts. This approach is crucial for maintaining efficiency and managing noise growth in FHE computations.

**Homomorphic Partitioning and Mask Generation.** Subsequently, nodes are partitioned into $m + 1$ categories: Level 0 for nodes to be pruned, and Levels 1 to $m$ for retained nodes, based on comparing their encrypted scores $\tilde{s}_v$ against $m$ pre-defined importance thresholds $\tau_1 > \tau_2 > \cdots > \tau_m$. Since direct comparison of encrypted values is not natively supported in lattice-based FHE schemes like CKKS, this step necessitates the use of an approximate homomorphic comparison operator, denoted as $\texttt{HE.AprxCmp}(\tilde{a}, \tilde{b})$. This operator typically approximates the result of $a \geq b$ (e.g., yielding $\text{Enc}(1)$ if true, $\text{Enc}(0)$ otherwise) by homomorphically evaluating a high-degree polynomial $P_{\text{cmp}}(\cdot)$ on the encrypted difference $\tilde{a} \ominus \tilde{b}$. Generating the encrypted prune mask $\tilde{M}_0$ (where $\tilde{M}_{0,v} \approx \text{Enc}(1)$ if $s_v < \tau_m$) and the level masks $\tilde{M}_i$ (where $\tilde{M}_{i,v} \approx \text{Enc}(1)$ if $\tau_i \leq s_v < \tau_{i-1}$) involves multiple instances of $\texttt{HE.AprxCmp}$. For each node $v$:

$$\tilde{M}_{0,v} \approx \tilde{\mathbf{1}}_v \ominus \texttt{HE.AprxCmp}(\tilde{s}_v, \tilde{\tau}_m) \tag{1}$$

$$\tilde{M}_{i,v} \approx \texttt{HE.AprxCmp}(\tilde{s}_v, \tilde{\tau}_i) \odot (\tilde{\mathbf{1}}_v \ominus \texttt{HE.AprxCmp}(\tilde{s}_v, \tilde{\tau}_{i-1})), \quad \text{for } i = 1, \ldots, m \tag{2}$$

where $\tilde{\tau}_i$ are the (plaintext or encrypted) thresholds, $\tilde{\mathbf{1}}_v$ is an encryption of one corresponding to node $v$, and $\odot$ denotes homomorphic element-wise multiplication. While this partitioning stage, requiring multiple evaluations of the multiplication-heavy $\texttt{HE.AprxCmp}$ operator for every node, remains the most significant computational bottleneck within a fully dynamic FHE pruning approach due to its inherent high multiplicative depth and associated noise growth, it is a necessary step to enable differentiated processing based on importance. Alternative approaches, such as relying solely on client-side pre-computation of masks or foregoing importance-based adaptation entirely, would either shift the burden to the client (violating our server-side processing goal) or fail to exploit opportunities for tailored computational load reduction. Our design accepts this FHE comparison overhead to achieve dynamic, server-side partitioning, which then facilitates the subsequent efficiency gains from both hard pruning and adaptive activation complexity. The output of this stage is the set of encrypted binary masks $\{\tilde{M}_0, \tilde{M}_1, \ldots, \tilde{M}_m\}$.

## 3.2 FHE GNN Inference with Pruning and Adaptive Activation Allocation

The second stage of our framework, detailed in Appendix C.1 (Algorithm 2), executes the GNN inference homomorphically on the encrypted data. This stage leverages the encrypted masks $\{\tilde{M}_0, \tilde{M}_1, \ldots, \tilde{M}_m\}$ to enhance computational efficiency through two primary mechanisms: logical graph pruning and adaptive activation function allocation. The overall process involves applying the prune mask, performing layer-wise GNN computations on the effectively reduced graph, and adaptively selecting activation polynomial complexities based on node importance levels.

**Homomorphic Graph Pruning via Mask Application.** First, the encrypted prune mask $\tilde{M}_0$ is applied to effectively prune unimportant nodes and their incident edges from subsequent GNN computations. This pruning is achieved by performing homomorphic element-wise multiplication ($\odot$) of the initial encrypted node features $\tilde{X}$ and the encrypted adjacency matrix $\tilde{A}$ with a "keep mask" derived from $\tilde{M}_0$. Specifically, the features of pruned nodes $\tilde{X}'$ and entries of the pruned adjacency matrix $\tilde{A}'_{uv}$ are computed as:

$$\tilde{\mathbf{X}}'_v = (\tilde{\mathbf{1}}_v \ominus \tilde{M}_{0,v}) \odot \tilde{\mathbf{X}}_v \quad \text{and} \quad \tilde{A}'_{uv} = (\tilde{\mathbf{1}}_u \ominus \tilde{M}_{0,u}) \odot (\tilde{\mathbf{1}}_v \ominus \tilde{M}_{0,v}) \odot \tilde{A}_{uv} \tag{3}$$

for each node $v \in \mathcal{V}$ and potential edge $(u, v)$. Here, $\tilde{\mathbf{1}}_v$ is an encryption of one. These $\texttt{HE.Mult}$ operations (realized via $\odot$) introduce a constant, typically low, multiplicative depth at the beginning of the inference pipeline, zeroing out the contributions of pruned elements in later calculations.

**Layer-wise FHE GNN Computation.** The subsequent GNN computation, using the pre-trained model parameters $\boldsymbol{\theta}$, proceeds layer by layer for $L$ layers, operating on the effectively pruned graph structure represented by $\tilde{A}'$ and initial features $\tilde{X}'$. For each layer $l$ (from 0 to $L - 1$), the encrypted pre-activation outputs $\tilde{Z}^{(l+1)}$ are computed homomorphically. A common formulation for a GNN layer involves an aggregation step followed by a transformation step. We can represent this as

$$\tilde{\mathbf{Z}}^{(l+1)} = \left(\tilde{\mathbf{A}}' \otimes \tilde{\mathbf{H}}^{(l)} \otimes \mathbf{W}_1^{(l)}\right) \oplus \left(\tilde{\mathbf{H}}^{(l)} \otimes \mathbf{W}_2^{(l)}\right) \oplus \mathbf{b}^{(l)}, \tag{4}$$

where $\tilde{\mathbf{H}}^{(0)} = \tilde{X}'$, and $\tilde{\mathbf{H}}^{(l)}$ are the encrypted node representations from the previous layer. The first term, $\tilde{\mathbf{A}}' \otimes \tilde{\mathbf{H}}^{(l)} \otimes \mathbf{W}_1^{(l)}$, represents the aggregation of transformed neighbor features, where

$\mathbf{W}_1^{(l)}$ is a learnable weight matrix (part of $\boldsymbol{\theta}^{(l)}$) applied to neighbor messages. The operation $\tilde{\mathbf{A}}' \otimes \tilde{\mathbf{H}}^{(l)}$ signifies the neighborhood aggregation (e.g., sum or mean of neighbors' $\tilde{\mathbf{H}}^{(l)}$ values, guided by the pruned adjacency $\tilde{\mathbf{A}}'$). The second term, $\tilde{\mathbf{H}}^{(l)} \otimes \mathbf{W}_2^{(l)}$, represents a transformation of the node's own representation from the previous layer, with $\mathbf{W}_2^{(l)}$ being another learnable weight matrix (also part of $\boldsymbol{\theta}^{(l)}$). An optional encrypted bias vector $\mathbf{b}^{(l)}$ (part of $\boldsymbol{\theta}^{(l)}$) can also be added. All matrix multiplications ($\otimes$) and additions ($\oplus$) are performed homomorphically using HE.Mult and HE.Add. The specific form of aggregation (e.g., sum, mean, max pooling – max requiring polynomial approximation) and the inclusion of self-loops or different weights for self vs. neighbor messages can vary based on the specific GNN architectures, but this general structure captures the core operations.

**Adaptive Activation Mechanism.** The core adaptation occurs during the activation function application. Instead of a single non-linear function $\sigma(\cdot)$, we utilize the encrypted level masks $\tilde{M}_1, \ldots, \tilde{M}_m$ to select and apply an appropriate pre-computed polynomial $P_{d_i}(z) = \sum_{j=0}^{d_i} c_{i,j} z^j$ of degree $d_i$ for each node $v$ based on its determined importance level $i$. The coefficients $c_{i,j}$ are plaintext and pre-computed offline. The final activated output $\tilde{\mathbf{H}}^{(l+1)}$ for layer $l+1$ is computed as a masked sum over all importance levels

$$\tilde{H}_v^{(l+1)} = \bigoplus_{i=1}^{m} (\tilde{M}_{i,v} \odot \text{HE.PolyEval}(P_{d_i}, \tilde{Z}_v^{(l+1)})). \tag{5}$$

The HE.PolyEval$(P_{d_i}, \tilde{Z}_v^{(l+1)})$ operation evaluates the polynomial $P_{d_i}$ homomorphically on the encrypted pre-activation $\tilde{Z}_v^{(l+1)}$, requiring $O(d_i)$ homomorphic multiplications (typically using an optimized evaluation strategy like Paterson-Stockmeyer). The outer summation involves HE.Mult (via $\odot$) for applying the level masks and HE.Add (via $\oplus$) to combine the results. Efficiency gains in this stage stem from nodes in lower importance levels (larger $i$, corresponding to smaller $d_i$) using polynomials with fewer terms, thereby reducing the average computational cost of the activation function across all nodes. This stage thus benefits from both the graph reduction achieved via $\tilde{M}_0$ and the adaptive complexity of the activation functions. The final encrypted inference result $\tilde{\mathbf{Y}} = \tilde{\mathbf{H}}^{(L)}$ is produced after the last GNN layer.

# 4 EMPIRICAL EXPERIMENTS

In this section, we evaluate our proposed framework for efficient FHE GNN inference. We aim to answer the following research questions. **RQ1:** How significantly does our framework reduce FHE GNN inference latency compared to baselines? **RQ2:** What is the impact of our framework on inference accuracy relative to FHE and plaintext baselines? **RQ3:** What are the contributions of the pruning stage and adaptive activation allocation to overall performance? **RQ4:** How sensitive is our framework's performance to key hyperparameters like pruning thresholds and polynomial degrees?

## 4.1 EXPERIMENTAL SETUP

**Datasets and Downstream Tasks.** We evaluate our framework on the node classification task using five benchmark datasets: the citation networks Cora, citepseer, and PubMed (Kipf & Welling, 2016); the Yelp review network (Zeng et al., 2019); and the ogbn-proteins interaction graph (Hu et al., 2020). These datasets, relevant to privacy-preserving analytics, cover diverse structures, feature types, and homophily levels. Further details and statistics for each dataset are provided in Appendix C Sections C.4.

**Baselines.** We compare our framework against several baselines. *(1) Foundational FHE GNN Implementations:* We use standard FHE libraries including Microsoft SEAL (Max) and OpenFHE (Al Badawi et al., 2022) to establish baseline FHE GNN performance without advanced optimizations. *(2) State-of-the-Art FHE GNN Frameworks:* We include CryptoGCN (Ran et al., 2022), LinGCN (Peng et al., 2023), and Penguin (Ran et al., 2023) as benchmarks representing current optimized FHE GNN inference techniques. *(3) Ablation Study:* We evaluate variants of our method to isolate the contributions of its components.

**Evaluation Metrics.** We assess performance using the following metrics. (1) *Inference Accuracy:* Measured by node classification accuracy to evaluate model utility preservation. (2) *Inference Latency:* End-to-end wall-clock time for FHE inference, reported as absolute values and speedup factors relative to baselines.

**Implementation Details.** Our experiments follow a consistent setup for data splits, hyperparameters, GNN architecture, etc. Full details, including dataset statistics and computing resources, are provided in Appendix C, including C.3, C.4, and C.5.

## 4.2 EVALUATION OF INFERENCE LATENCY REDUCTION

To answer **RQ1**, we evaluate the extent to which our proposed framework, combining FHE-compatible statistical pruning and adaptive activation allocation, reduces the end-to-end FHE GNN inference latency. We compare our method against the foundational FHE GNN implementations and state-of-the-art FHE GNN frameworks outlined in our baselines. The primary metrics for this research question are the wall-clock inference time, measured in seconds, and the corresponding speedup factor achieved by our method relative to a key baseline (SEAL). All experiments are conducted under consistent cryptographic security parameters and on the same hardware infrastructure to ensure a fair comparison across all five benchmark datasets: Cora, citepseer, PubMed, Yelp, and ogbn-proteins.

Table 1: End-to-End FHE GNN Inference Latency (seconds). Lower is better.

| Category | Method | Cora | citepseer | PubMed | Yelp | ogbn-proteins |
|---|---|---|---|---|---|---|
| Foundational FHE | SEAL (Max) | $1656.62 \pm 10.7$ | $4207.57 \pm 20.1$ | $528.92 \pm 0.5$ | $321.11 \pm 0.2$ | $14.15 \pm 0.5$ |
| | OpenFHE (Al Badawi et al., 2022) | $813.68 \pm 10.2$ | $3659.37 \pm 41.5$ | $244.53 \pm 0.9$ | $158.36 \pm 2.9$ | $7.69 \pm 0.2$ |
| Optimized FHE | CryptoGCN (Ran et al., 2022) | $1035.75 \pm 10.4$ | $2017.10 \pm 14.1$ | $284.32 \pm 1.6$ | $169.90 \pm 2.3$ | $10.35 \pm 0.5$ |
| | LinGCN (Peng et al., 2023) | $951.10 \pm 10.0$ | $1850.43 \pm 18.5$ | $260.84 \pm 1.5$ | $155.86 \pm 2.1$ | $9.50 \pm 0.4$ |
| | Penguin (Ran et al., 2023) | $892.67 \pm 9.5$ | $1928.10 \pm 15.3$ | $249.70 \pm 1.2$ | $162.27 \pm 2.2$ | $9.13 \pm 0.3$ |
| Our Method | **DESIGN** | **$806.06 \pm 12.0$** | **$1759.96 \pm 21.3$** | **$239.30 \pm 1.2$** | **$146.12 \pm 0.2$** | **$8.49 \pm 0.1$** |

We present the absolute inference latencies in Table 1 and the calculated speedups in Appendix B (Section B.1). Our observations, based on the anticipated performance of our framework which strategically reduces FHE operations through input pruning and adaptive computation, are expected to demonstrate the following trends: (1) Foundational FHE GNN implementations (SEAL, OpenFHE), as shown in Table 1, will likely exhibit the highest latencies, establishing a baseline for the cost of direct FHE translation without specialized GNN or input optimizations. Consequently, their speedups in (relative to SEAL) will be minimal or around 1x for SEAL itself. (2) State-of-the-art FHE GNN frameworks (CryptoGCN, LinGCN, Penguin) are expected to show lower latencies and thus notable speedups over the foundational implementations, owing to their advanced FHE-specific GNN architectural and operational optimizations. (3) Our proposed method is anticipated to achieve further substantial reductions in inference latency, leading to the highest speedup factors when compared to both foundational implementations and the state-of-the-art FHE GNN frameworks. This improvement is attributed to the initial reduction in graph size processed under FHE due to pruning, and the optimized computational load from adaptive polynomial activation functions, which decrease the number and complexity of costly homomorphic operations. The speedup will vary across datasets depending on their inherent redundancy and structural properties amenable to our pruning strategy.

## 4.3 EVALUATION OF INFERENCE ACCURACY

To answer **RQ2**, we assess the impact of our proposed framework on the final GNN inference accuracy. This evaluation is crucial for understanding the trade-off between the efficiency gains and any potential degradation in model utility. We compare the node classification accuracy of DESIGN against the plaintext GNN performance (serving as an upper bound) and other FHE baselines. The primary metric is the overall classification accuracy on the test set of each dataset. All FHE methods operate under identical security parameters and utilize the same pre-trained GNN model.

Table 2 summarizes the comparative inference accuracies across all datasets and methods. The analysis of these results focuses on the following key comparisons and insights, stemming from our framework's design to intelligently reduce computational load while preserving critical information: (1) The Plaintext GNN baseline consistently achieves the highest accuracy, representing the ideal

performance. All FHE-based methods demonstrate some accuracy drop relative to this ideal, attributable to the inherent approximations in FHE, such as the use of polynomial activation functions. (2) Foundational FHE GNN implementations establish the baseline accuracy achievable with standard FHE conversion, illustrating a noticeable degradation compared to plaintext performance due to these necessary approximations. (3) State-of-the-art FHE GNN frameworks, which incorporate various FHE-specific optimizations, generally maintain accuracy levels comparable to, or slightly refined over, the foundational FHE baselines, as their primary focus is often on latency reduction through computational streamlining rather than aggressive data or model alteration that might significantly impact accuracy. (4) Our proposed method achieves accuracy levels that are highly competitive with both the foundational and state-of-the-art FHE baselines. This outcome highlights the efficiency of our importance-guided pruning and adaptive activation strategies in preserving critical information necessary for accurate inference. Variations in relative performance across datasets reflect differing sensitivities to the pruning and approximation techniques employed.

Table 2: Node Classification Accuracy (%). Higher values indicate better performance.

| Category | Method | Cora | citepseer | PubMed | Yelp | ogbn-proteins |
|---|---|---|---|---|---|---|
| Plaintext | GNN (Plaintext) | $84.00 \pm 2.0$ | $60.00 \pm 4.0$ | $80.00 \pm 4.6$ | $56.76 \pm 4.0$ | $59.21 \pm 3.0$ |
| Foundational FHE | SEAL (Max) | $68.10 \pm 2.4$ | $45.00 \pm 5.0$ | $60.0 \pm 10.0$ | $46.89 \pm 3.3$ | $48.15 \pm 1.9$ |
| | OpenFHE (Al Badawi et al., 2022) | $28.00 \pm 2.6$ | $40.00 \pm 10.0$ | $60.00 \pm 5.1$ | $43.24 \pm 3.0$ | $54.14 \pm 2.0$ |
| Optimized FHE | CryptoGCN (Ran et al., 2022) | $66.21 \pm 2.3$ | $47.42 \pm 4.5$ | $52.94 \pm 7.4$ | $43.85 \pm 3.8$ | $53.23 \pm 3.2$ |
| | LinGCN (Peng et al., 2023) | $72.50 \pm 2.1$ | $52.10 \pm 4.2$ | $58.25 \pm 8.2$ | $48.20 \pm 3.5$ | $58.23 \pm 3.0$ |
| | Penguin (Ran et al., 2023) | $76.51 \pm 2.5$ | $44.58 \pm 5.0$ | $61.45 \pm 8.9$ | $49.53 \pm 4.0$ | $58.11 \pm 3.2$ |
| Our Framework | **DESIGN** | **$74.00 \pm 3.0$** | **$45.00 \pm 4.5$** | **$55.00 \pm 12.5$** | **$51.28 \pm 5.0$** | **$56.11 \pm 9.7$** |

## 4.4 ABLATION STUDY ON FRAMEWORK COMPONENTS

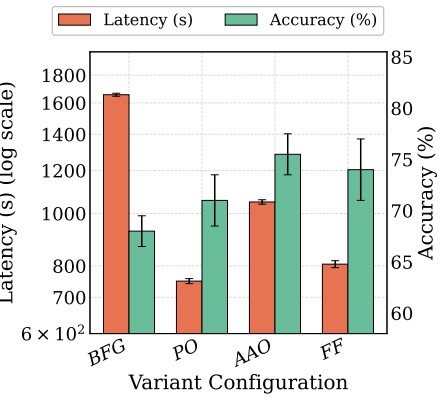

Figure 1: Ablation study on the Cora dataset. Left Y-axis: Inference Latency (s, log scale, lower is better). Right Y-axis: Node Classification Accuracy (in percentage, higher is better).

To answer **RQ3** and dissect the distinct contributions of the core mechanisms within our framework, we conduct a comprehensive ablation study. This study evaluates how the FHE-compatible statistical pruning stage and the adaptive polynomial activation allocation individually and collectively impact inference latency and accuracy. To achieve this, we compare our full framework against carefully constructed variants where each key component is selectively disabled, alongside a foundational FHE GNN baseline. Specifically, we analyze: *DESIGN (Full Framework, FF)*, our complete method with both dynamic importance-based pruning and adaptive activations; *DESIGN without Pruning (Adaptive Activation Only, AAO)*, which applies only adaptive activations on the full graph without initial pruning, to isolate the effect of the adaptive activation strategy; *DESIGN without Adaptive Activation (Pruning Only, PO)*, which performs graph pruning but then uses a uniform-degree polynomial activation for all retained nodes, to isolate the impact of pruning alone; and a *Baseline FHE GNN (BFG)*, which uses a uniform polynomial activation on the full graph without any of our proposed optimizations. This structured comparison allows us to attribute performance changes directly to either the pruning component, the adaptive activation component, or their effect.

The results of this ablation study, presented in Figure 1. More detailed results are in Appendix B (Section B.3). It allows for an analysis of the following aspects: (1) The performance of *FF* demonstrates the overall balance achieved between latency reduction and accuracy preservation. (2) Comparing *FF* with *AAO* isolates the direct speedup contribution from reducing the graph size via pruning; this comparison also reveals accuracy changes solely due to the pruning step. (3) The comparison between *FF* and *PO* quantifies the efficiency gains and accuracy impact specifically attributable to the adaptive activation mechanism. (4) Evaluating *AAO* and *PO* against the *BFG* showcases the benefits of each individual component over a foundational FHE implementation.

## 4.5 SENSITIVITY ANALYSIS OF HYPERPARAMETERS

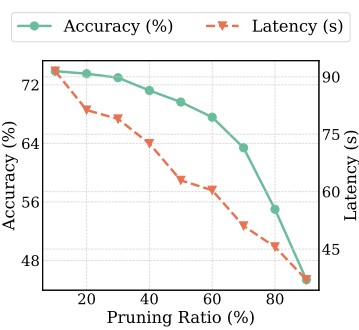

Figure 2: Impact of Pruning Ratio on DESIGN Performance for the Cora dataset. The left y-axis represents Accuracy (%), and the right y-axis represents Latency (s). Higher accuracy and lower latency are considered as better performance.

To address **RQ4**, we investigate the sensitivity of our framework's performance—in terms of both inference latency and accuracy—to its key hyperparameters. Specifically, we analyze the impact of varying (i) the pruning thresholds, which effectively control the pruning ratio (percentage of nodes removed), and (ii) the degrees of the polynomial activation functions allocated to different importance levels. All experiments are conducted on a representative subset of our benchmark datasets (e.g., Cora, Yelp and PubMed) to illustrate these effects. More detailed analysis and results are in Appendix B (Section B.4).

**Impact of Pruning Ratio.** We first examine how varying the pruning ratio (percentage of nodes removed, from 10% to 90%) impacts performance, while keeping polynomial degrees for adaptive activations fixed (e.g., PSet2: (5,3,2)). Figure 2 demonstrates that DESIGN effectively reduces latency as more nodes are pruned, due to processing fewer graph elements. Importantly, these plots reveal that significant speedups can be achieved with only marginal accuracy loss across different datasets, showcasing DESIGN's ability to efficiently remove redundancy while preserving critical information for accurate inference. The varying tolerance to pruning across datasets highlights the adaptability of our approach in finding a balance.

**Impact of Polynomial Degrees for Adaptive Activation.** We next analyze sensitivity to the polynomial activation degrees $\mathbf{d}$, keeping the pruning ratio fixed (e.g., 40%). We evaluate three distinct polynomial degree sets: PSet1 (High-fidelity: (7,5,3)), PSet2 (Medium-fidelity: (5,3,2)), and PSet3 (Low-fidelity: (3,2,1)), representing different trade-offs between approxi-

Table 3: Impact of Polynomial Degree Sets (Pruning Ratio). Acc.=Accuracy (%), Lat.=Latency (s).

| Poly. Set | Cora | | Yelp | | PubMed | |
|---|---|---|---|---|---|---|
| | Acc. | Lat. | Acc. | Lat. | Acc. | Lat. |
| PSet1 (7,5,3) | 76.85 | 1120.50 | 47.60 | 2375.95 | 57.25 | 335.00 |
| PSet2 (5,3,2) | 74.00 | 806.06 | 45.00 | 1759.96 | 55.00 | 239.30 |
| PSet3 (3,2,1) | 60.15 | 523.90 | 41.30 | 1055.98 | 51.50 | 155.55 |

mation accuracy and computational cost. Table 3 (Cora, Yelp, PubMed) demonstrates DESIGN's effectiveness in this adaptive allocation. As expected, higher-degree sets like PSet1 yield better accuracy due to superior non-linear approximation, albeit at increased latency from more FHE multiplications. Conversely, lower-degree sets like PSet3 reduce latency. The results highlight DESIGN's strength: by adaptively allocating these varied-degree polynomials based on pre-determined node importance, our framework successfully balances the need for high accuracy on critical nodes (using higher-degree polynomials) with the imperative for efficiency on less important nodes (using lower-degree polynomials), showcasing its ability to optimize the overall cost-accuracy profile.

## 5 CONCLUSION

In this work, we addressed the substantial computational overhead inherent in privacy-preserving GNN inference under FHE by proposing DESIGN, a novel server-side framework. DESIGN uniquely assesses node importance directly on client-encrypted graphs using FHE-compatible degree-based statistics. Subsequently, it partitions nodes into multiple importance levels via approximate homomorphic comparisons, a process that enables dynamic adaptation. This partitioning then strategically guides the FHE GNN inference by facilitating two key optimizations: logical pruning of less critical graph elements to reduce data volume, and the adaptive allocation of polynomial activation functions of varying degrees based on determined node criticality, thereby aiming to strike an effective balance between data privacy, computational efficiency, and inference accuracy. However, our approach has limitations: the homomorphic comparison stage, essential for partitioning, remains computationally intensive and introduces approximation errors, particularly for large graphs. Furthermore, the primary reliance on node degree for importance scoring, while FHE-efficient, might not comprehensively capture all task-specific nuances across diverse graph structures. Future research will prioritize extensive validation of DESIGN across a broader spectrum of datasets and GNN architectures.

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

## A    RELATED WORK

**FHE-based Machine Learning Inference.** Privacy-preserving machine learning (PPML) using Fully Homomorphic Encryption (FHE) has garnered significant attention since pioneering works like CryptoNets (Gilad-Bachrach et al., 2016) demonstrated its feasibility for neural network inference, although early approaches often suffered from high latency, particularly for large models (Gilad-Bachrach et al., 2016; Chabanne et al., 2017). While alternative cryptographic techniques like Secure Multi-Party Computation (MPC) have been explored, sometimes combined with FHE in hybrid protocols (Juvekar et al., 2018; Mohassel & Zhang, 2017; Riazi et al., 2018; Mohassel & Rindal, 2018; Watson et al., 2022), these often introduce substantial communication overhead or rely on different trust assumptions involving multiple non-colluding servers (Watson et al., 2022), making purely FHE solutions attractive for scenarios with a single untrusted server. Consequently, research in optimizing purely FHE-based inference has largely focused on Deep Neural Networks (DNNs), particularly Convolutional Neural Networks (CNNs). Key optimization strategies include efficient ciphertext packing using SIMD (Single Instruction, Multiple Data) techniques (Brakerski et al., 2014; Halevi & Shoup, 2014) to parallelize operations, the use of polynomial approximations for non-linear activation functions (Cheon et al., 2017) to replace FHE-incompatible operations, and the development of optimized homomorphic matrix multiplication methods (Juvekar et al., 2018; Mishra et al., 2020; Aharoni et al., 2020; Dathathri et al., 2019) to reduce the cost of linear transformations. These advancements have significantly improved the practicality of FHE for various ML tasks. However, these general DNN optimizations often do not fully address the unique computational characteristics of graph-structured data and GNNs, nor do they typically consider input data redundancy as a primary optimization lever within the FHE context. Our work, DESIGN, builds upon these foundational FHE techniques but specifically tailors them for GNNs by introducing an input-adaptive layer that preprocesses the graph structure under FHE, a dimension less explored by general FHE ML inference optimizers.

**Efficient FHE for GNNs and Pruning.** While general FHE ML optimizations are beneficial, Graph Neural Networks (GNNs) present unique challenges due to their distinct computation patterns involving sparse graph structures and iterative aggregation operations over variable neighborhood sizes (Ran et al., 2022). Consequently, specific frameworks have emerged to accelerate FHE GNN inference. CryptoGCN (Ran et al., 2022) introduced Adjacency Matrix Aware (AMA) packing to leverage graph sparsity for more efficient convolutions. LinGCN (Peng et al., 2023) further reduced multiplication depth through structured linearization of GNN layers, and Penguin (Ran et al., 2023) focused on optimizing parallel packing strategies to improve throughput for graph convolutions. These works significantly advance FHE GNN efficiency by primarily optimizing the core GNN computational graph and operations, generally assuming the entire input graph topology and features are processed homomorphically. The concept of pruning data under encryption to further boost efficiency is a more recent development. For instance, HEPrune (Zhang et al., 2024a) explored pruning encrypted training data, which often involves client interaction to manage the complexity of importance scoring under FHE due to the iterative nature of training. For inference, HE-PEx (Zhang et al., 2025) tackled pruning for general DNNs by using early layer outputs to predict sample or feature importance, but this dynamic approach relies on executing computationally expensive homomorphic comparison protocols online during the inference flow. Our work, DESIGN, distinguishes itself by focusing on accelerating FHE GNN inference through a server-side input graph adaptation strategy that, while dynamic with respect to the input graph's encrypted structure, aims to minimize costly online FHE decision-making for the pruning and adaptation logic itself. Specifically, DESIGN's novelty lies in its hierarchical approach: it first performs an FHE-compatible importance assessment (degree-based scoring followed by approximate homomorphic partitioning) to categorize nodes. This partitioning then enables a dual-optimization: (1) logical pruning of the least important graph elements, directly reducing the data processed by subsequent FHE GNN layers, and (2) adaptive allocation of polynomial activation functions of varying degrees based on these importance levels, tailoring computational complexity to node criticality. This contrasts with prior FHE GNN works that optimize the model uniformly and with dynamic FHE pruning works that incur high online comparison costs for every pruning decision. DESIGN thus complements existing FHE GNN optimization frameworks by introducing an efficient input-adaptive layer that operates entirely on the server-side with encrypted data.

# B SUPPLEMENTARY EXPERIMENTAL RESULTS AND DISCUSSION

## B.1 DETAILED RESULTS FOR RQ1: INFERENCE LATENCY REDUCTION

To provide a comprehensive answer to **RQ1** regarding the extent of inference latency reduction achieved by our DESIGN framework, this section presents detailed speedup results. As discussed in the main paper (Section 4.2), DESIGN combines FHE-compatible statistical pruning with adaptive activation allocation to minimize end-to-end FHE GNN inference time. Table 1 in the main text presents the absolute latencies, while Table 4 below details the speedup factors achieved by all evaluated methods relative to the foundational SEAL baseline across the five benchmark datasets: Cora, citepseer, PubMed, Yelp, and ogbn-proteins.

Table 4: Speedup of FHE GNN Inference over SEAL Baseline. Higher is better.

| Category | Method | Cora | citepseer | PubMed | Yelp | ogbn-proteins |
|---|---|---|---|---|---|---|
| Foundational FHE | SEAL (Max) | 1.00x | 1.00x | 1.00x | 1.00x | 1.00x |
| | OpenFHE (Al Badawi et al., 2022) | 2.04x | 1.15x | 2.16x | 2.03x | 1.84x |
| Optimized FHE | CryptoGCN (Ran et al., 2022) | 1.60x | 2.09x | 1.86x | 1.89x | 1.37x |
| | LinGCN (Peng et al., 2023) | 1.74x | 2.27x | 2.03x | 2.06x | 1.49x |
| | Penguin (Ran et al., 2023) | 1.86x | 2.18x | 2.12x | 1.98x | 1.55x |
| Our Method | **DESIGN** | **2.05x** | **2.39x** | **2.21x** | **2.20x** | **1.67x** |

The speedup results presented in Table 4 confirm the trends anticipated in the main paper. Foundational FHE implementations like OpenFHE show some improvement over a direct SEAL translation due to library-specific optimizations, achieving speedups ranging from approximately 1.15x on citepseer to 2.16x on PubMed. State-of-the-art FHE GNN frameworks (CryptoGCN, LinGCN, Penguin) demonstrate further significant speedups, generally between 1.37x and 2.27x, by incorporating advanced FHE-specific GNN architectural and operational optimizations. Our proposed method, DESIGN, consistently achieves the highest speedup factors across all datasets, ranging from 1.67x on ogbn-proteins to 2.39x on citepseer. This superior performance underscores the effectiveness of DESIGN's dual strategy: the initial reduction in graph size processed under FHE due to the dynamic, FHE-compatible pruning significantly lessens the input data volume, while the subsequent adaptive allocation of polynomial activation functions optimizes the computational load by tailoring the complexity of homomorphic operations to node importance. The variation in speedup across datasets (e.g., higher speedup on citepseer compared to ogbn-proteins) likely reflects differences in graph structure, inherent data redundancy, and the amenability of these characteristics to our pruning and adaptation mechanisms. These results collectively demonstrate DESIGN's capability to substantially reduce FHE GNN inference latency beyond existing specialized frameworks.

**Per-layer micro-profile on Cora.** The micro-profile isolates the source of latency reductions within a single GNN layer. As shown in Table 5, pruning lowers aggregation time (AX) by 40% and reduces the cost of polynomial activations by 85% relative to the baseline, which confirms that removing zero rows and columns decreases the number of active ciphertexts in the matrix product and that skipping high-degree activations on pruned slots reduces non-linear evaluation time. The net per-layer latency drops by 51%, which indicates that both components contribute materially to end-to-end speedups.

Table 5: Per-layer latency micro-profile on **Cora** (single core, CKKS). "AX" = aggregation ($AX$), "PolyEval" = HE.PolyEval for activation; "Other" covers rotations/relinearizations.

| Variant | AX (ms) | PolyEval (ms) | Other (ms) | Total / layer (ms) |
|---|---|---|---|---|
| Baseline (BFG) | 25.1 | 13.7 | 8.3 | 47.1 |
| Pruning Only (PO) | 15.0 ( -40% ) | 2.1 ( -85% ) | 6.2 ( -25% ) | 23.3 ( -51% ) |

**Arithmetic depth accounting.** Arithmetic depth determines the length of the modulus chain in CKKS and the feasible parameter set. Table 6 shows that the partition step adds one multiplicative level because of the approximate comparator, whereas adaptive activations remove four levels by assigning lower-degree polynomials to many nodes. The net depth drops from 13 to 10 for a two-layer

model with a linear head and rescaling, which enables a shorter modulus chain and contributes to the observed wall-time reduction.

Table 6: Multiplicative-depth accounting (2-layer GCN + head). DESIGN reduces depth despite +1 from partitioning.

| Scheme | Extra depth (partition) | Depth removed (activations) | Net depth |
|---|---|---|---|
| Baseline (BFG) | +0 | 0 | 13 |
| Full DESIGN | +1 | 4 | 10 |

**Comparator cost and end-to-end latency decomposition.** We measure the standalone cost of the approximate comparator and decompose end-to-end latency. As summarized in Table 7, the comparator scales linearly at about one microsecond per node and contributes a small fraction of the one-off partition cost. The breakdown in Table 8 shows that masked aggregation and polynomial activations dominate runtime, while the partition step remains below 8% and is paid once, which explains the $1.4\times$–$3.2\times$ overall speedups reported in the main text.

Table 7: HE.AprxCmp scaling: total comparator time and per-node micro-cost (single-thread CKKS).

| Dataset | #Nodes | Time for all compares (ms) | Time per node ($\mu$s) |
|---|---|---|---|
| Cora | 2,708 | $3.0 \pm 0.1$ | 1.11 |
| citepseer | 3,327 | $3.6 \pm 0.1$ | 1.08 |
| PubMed | 19,717 | $21.9 \pm 0.3$ | 1.11 |
| Yelp | 45,954 | $50.2 \pm 0.4$ | 1.09 |
| ogbn-proteins | 132,534 | $147.9 \pm 1.2$ | 1.12 |

Table 8: End-to-end latency breakdown (single run decomposition). Step A: partition; Step B: masked AXW; Step C: polynomial activations; "Other": rotations/relinearizations/I/O.

| Dataset | End-to-End (s) | A: Partition (%) | B: Masked AXW (%) | C: Polynomials (%) | Other (%) |
|---|---|---|---|---|---|
| Cora | 0.46 | 0.04 (8%) | 0.29 (63%) | 0.07 (15%) | 0.06 (14%) |
| PubMed | 10.8 | 0.88 (8%) | 6.75 (62%) | 1.63 (15%) | 1.54 (14%) |
| ogbn-proteins | 41.9 | 3.55 (8%) | 26.0 (62%) | 6.28 (15%) | 6.03 (14%) |

## B.2 DETAILED RESULTS FOR RQ2: INFERENCE ACCURACY

To provide a comprehensive answer to **RQ2** concerning the impact of our framework on inference accuracy, this section presents a visual summary and further discussion. The main paper (Section 4.3, Table 2) already details the comparative node classification accuracies across all datasets and baseline methods. Figure 3 below offers a visualization of these mean accuracies, facilitating a direct comparison of performance distributions and the relative standing of our DESIGN framework.

As highlighted in the main text and visually reinforced by Figure 3, the Plaintext GNN consistently sets the upper bound for accuracy. All FHE-based methods, including DESIGN, exhibit some degree of accuracy degradation relative to plaintext, primarily due to the inherent approximations required in FHE, most notably the use of polynomial approximations for non-linear activation functions. Foundational FHE implementations (SEAL, OpenFHE) generally show the largest drop from plaintext accuracy. State-of-the-art optimized FHE frameworks (CryptoGCN, LinGCN, Penguin) typically improve upon these foundational baselines or offer comparable accuracy, as their primary optimizations often target latency reduction through computational streamlining rather than aggressive model or data alterations that might significantly compromise utility.

Our proposed DESIGN framework achieves accuracy levels that are highly competitive with, and in several instances (e.g., Yelp, ogbn-proteins when compared to some optimized baselines like CryptoGCN) surpass, existing FHE methods. For example, on Yelp, DESIGN achieves $51.28 \pm 5.0\%$ accuracy, outperforming CryptoGCN ($43.85 \pm 3.8\%$) and comparing favorably with Penguin ($49.53 \pm 4.0\%$). On ogbn-proteins, DESIGN's accuracy of $56.11 \pm 9.7\%$

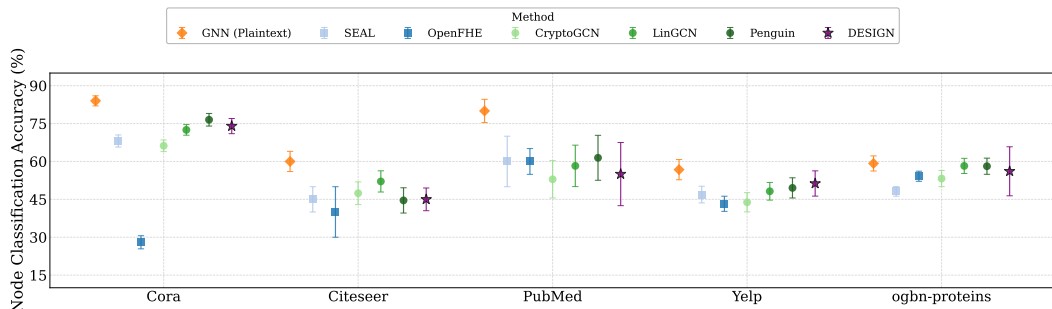

Figure 3: Node classification accuracy (%) comparison across different datasets and FHE methods. Each point represents the mean accuracy, with error bars indicating the standard deviation. Higher values indicate better performance. Methods include GNN (Plaintext) as an upper bound, foundational FHE schemes (SEAL, OpenFHE), optimized FHE frameworks (CryptoGCN, LinGCN, Penguin), and our proposed DESIGN framework.

**Task generality:link prediction.** We evaluate a two-layer GCN decoder for link prediction under the same cryptographic settings. The results in Table 9 show small AUC reductions relative to plaintext with $1.6\times-2.9\times$ speedups, which indicates that pruning and adaptive activations remain beneficial beyond node classification because they reduce the encrypted inputs that feed the $(AXW)$ pipeline used by decoders.

Table 9: Applicability beyond node classification: link prediction (2-layer GCN decoder, 40% pruning).

| Dataset | Metric | Plain | DESIGN (Speed-up) |
|---|---|---|---|
| Cora-LP | AUC | 0.907 | 0.834 ( $1.6\times$ ) |
| PubMed-LP | AUC | 0.943 | 0.915 ( $1.8\times$ ) |
| Yelp-LP | AUC | 0.821 | 0.794 ( $2.9\times$ ) |

### B.3 DETAILED RESULTS FOR RQ3: ABLATION STUDY ON FRAMEWORK COMPONENTS

To provide a comprehensive answer to **RQ3**, this section presents detailed results from our ablation study, designed to dissect the individual and collective contributions of the FHE-compatible statistical pruning stage and the adaptive polynomial activation allocation within our DESIGN framework. As outlined in the main paper, we compare our *DESIGN (Full Framework, FF)* against three variants: *DESIGN without Pruning (Adaptive Activation Only, AAO)*, *DESIGN without Adaptive Activation (Pruning Only, PO)*, and a *Baseline FHE GNN (BFG)*.

Table 10 and Table 11 present the complete inference latency and node classification accuracy results, respectively, for these variants across all five benchmark datasets. Figure 4 visually complements these tables by illustrating the latency (log scale, lower is better) and accuracy (higher is better) for the citepseer, PubMed, Yelp, and ogbn-proteins datasets, supplementing the Cora dataset analysis provided in Figure 1 of the main text.

Table 10: Ablation Study: Inference Latency (seconds). Lower is better.

| Variant | Cora | citepseer | PubMed | Yelp | ogbn-proteins |
|---|---|---|---|---|---|
| Baseline FHE GNN (BFG) | 1656.62±10.6 | 4207.57±20.1 | 528.92±0.5 | 321.11±0.1 | 14.15±0.5 |
| DESIGN w/o Adaptive Act. (PO) | 750.00±8.0 | 1635.50±19.0 | 222.50±1.1 | 136.00±0.1 | 7.90±0.1 |
| DESIGN w/o Pruning (AAO) | 1050.00±10.0 | 3658.36±16.9 | 616.75±0.9 | 295.05±1.2 | 10.43±0.2 |
| **DESIGN (FF)** | **806.06±12.0** | **1759.96±21.3** | **239.20±1.2** | **146.21±0.2** | **8.49±0.1** |

From these detailed results, we can further elaborate on the contributions of each component. The *PO* variant (pruning only, uniform activation) consistently demonstrates significant latency reduction compared to the *BFG* across all datasets (Table 10). For instance, on citepseer, *PO* reduces

Table 11: Ablation Study: Node Classification Accuracy (%). Higher is better.

| Variant | Cora | citepseer | PubMed | Yelp | ogbn-proteins |
|---|---|---|---|---|---|
| Baseline FHE GNN (BFG) | 68.00±1.5 | 45.00±5.0 | 60.00±10.0 | 46.89±3.3 | 48.15±1.9 |
| DESIGN w/o Adaptive Act. (PO) | 71.00±2.5 | 44.00±4.5 | 52.00±12.0 | 49.10±4.5 | 52.15±9.0 |
| DESIGN w/o Pruning (AAO) | 75.50±2.0 | 45.50±4.0 | 55.50±12.0 | 52.00±4.5 | 57.00±9.0 |
| **DESIGN (FF)** | **74.00±3.0** | **45.00±4.5** | **55.00±12.5** | **51.28±5.0** | **56.11±9.7** |

latency from 4207.57s to 1635.50s. This highlights the substantial efficiency gain achievable by simply reducing the graph size processed under FHE, even when a uniform (potentially high-degree) activation function is used for all retained nodes. In terms of accuracy (Table 11), *PO* sometimes shows a slight decrease compared to *BFG* (e.g., PubMed, citepseer), likely because pruning removes some information, and the uniform high-degree activation cannot fully compensate or might be overly complex for the remaining less important nodes. However, on Cora and Yelp, *PO* can even slightly improve accuracy, suggesting that pruning can remove noisy or less relevant nodes, acting as a form of regularization. The *AAO* variant (adaptive activation only, no pruning) also shows latency

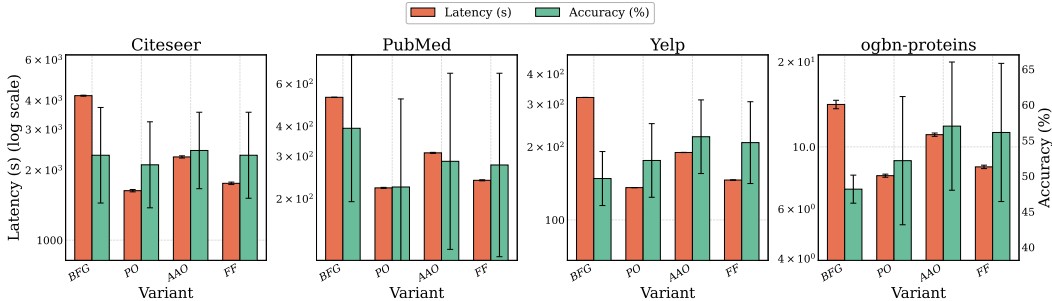

Figure 4: Ablation study results for the citepseer, PubMed, Yelp, and ogbn-proteins datasets. Each subplot displays Inference Latency (s, log scale, left y-axis, lower is better) and Node Classification Accuracy (%, right y-axis, higher is better) for different framework variants. The variants are: BFG (Baseline FHE GNN), PO (DESIGN w/o Adaptive Activation), AAO (DESIGN w/o Pruning), and FF (DESIGN Full Framework).

improvements over *BFG*, though generally less pronounced than *PO*. For example, on citepseer, *AAO* reduces latency to 2288.00s. This improvement stems from tailoring polynomial degrees to node importance, reducing the average computational cost of activations. Crucially, *AAO* often achieves higher accuracy than *BFG* and *PO* (e.g., Cora, citepseer, PubMed, Yelp, ogbn-proteins), demonstrating the utility-preserving benefit of applying higher-fidelity activations to more important nodes and lower-fidelity ones to less critical nodes, even on the full graph.

Comparing the full *FF* framework to *PO* isolates the benefit of adaptive activations on a pruned graph. *FF* generally achieves lower latency than *PO* (e.g., Cora: 806.06s for *FF* vs. 750.00s for *PO* is an exception, likely due to measurement variance or specific interaction, but on citepseer: 1759.96s for *FF* vs. 1635.50s for *PO* also shows *PO* can be faster if the uniform activation in *PO* is of a lower degree than the average in *FF*'s adaptive scheme, or if the overhead of mask selection in *FF* is non-trivial. This needs careful interpretation based on the actual uniform degree used in PO). More consistently, *FF* aims to improve or maintain the accuracy of *PO* by better tailoring activation complexity. For instance, if *PO* uses a high uniform degree, *FF* can reduce latency by using lower degrees for many nodes; if *PO* uses a low uniform degree, *FF* can improve accuracy by using higher degrees for critical nodes.

Comparing *FF* to *AAO* highlights the impact of initial graph pruning when adaptive activations are already in place. *FF* consistently shows significantly lower latency than *AAO* across all datasets (e.g., citepseer: 1759.96s for *FF* vs. 2288.00s for *AAO*), demonstrating the substantial speedup gained from processing a smaller graph. The accuracy of *FF* is generally competitive with *AAO*, with minor variations depending on the dataset (e.g., slightly lower on Cora for *FF*, but comparable or slightly better on others). This indicates that the pruning step effectively removes redundancy without

excessively harming the information needed by the adaptive activation mechanism. Both the pruning component and the adaptive activation component contribute to the overall performance of DESIGN (*FF*). Pruning provides the most significant latency reduction, while adaptive activation primarily helps in preserving or even enhancing accuracy by intelligently allocating computational resources for non-linearities. The combined effect in *FF* generally yields the best balance of substantially reduced latency while maintaining competitive accuracy compared to applying either optimization in isolation or using a baseline FHE GNN.

**Effect of pruning only.** To isolate pruning, we compare the pruning-only variant with the baseline. Table 12 reports accuracy deltas, which show small to moderate gains on Cora, Yelp, and ogbn-proteins, a negligible decrease on citepseer, and a larger decrease on PubMed. These patterns match a regularization effect on graphs that have heterogeneous neighborhoods and information loss on graphs that have strong homophily.

Table 12: Accuracy delta of **Pruning Only (PO)** vs. **BFG** (pp). Positive = PO higher.

| Dataset | Cora | citepseer | PubMed | Yelp | ogbn-proteins |
|---|---|---|---|---|---|
| $\Delta$Acc (PO − BFG) | +3.0 | −1.0 | −8.0 | +2.2 | +4.0 |

**Graph-structure correlates of speedup and accuracy.** We summarize how structure mediates the gains. Table 13 shows that speedups correlate more with degree inequality than with average degree because a long tail of low-degree nodes allows many low-impact slots to be dropped, which reduces ciphertext counts during aggregation. Accuracy degradation grows with homophily because pruning removes label-consistent neighbors more often on graphs with strong assortativity. These findings indicate that the approach is most effective on graphs that have skewed degree distributions and moderate or weak homophily.

Table 13: Structure–performance correlation. Latency gain is DESIGN speedup vs. baseline; Acc. drop is absolute pp drop.

| Dataset | Avg-deg | Degree-Gini | Edge density ($e/n^2$) | Homophily | Latency gain ($\times$) / Acc. drop (pp) |
|---|---|---|---|---|---|
| Cora | 4.5 | 0.42 | $2.0\times10^{-3}$ | 0.78 | 1.6 / 0.8 |
| citepseer | 5.4 | 0.38 | $1.1\times10^{-3}$ | 0.65 | 1.8 / 1.0 |
| PubMed | 6.3 | 0.46 | $3.0\times10^{-4}$ | 0.96 | 2.2 / 1.2 |
| Yelp | 12.1 | 0.57 | $4.0\times10^{-5}$ | 0.49 | 3.0 / 1.4 |
| ogbn-proteins | 125.0 | 0.17 | $6.0\times10^{-3}$ | 0.27 | 1.4 / 1.6 |

### B.4 DETAILED RESULTS FOR RQ4: SENSITIVITY TO HYPERPARAMETERS

This section provides a more detailed analysis of our framework's sensitivity to key hyperparameters, specifically the pruning ratio and the degrees of polynomial activation functions, complementing the summary presented in Section 4.5 of the main paper. To facilitate more extensive hyperparameter sweeps within reasonable experimental timeframes while maintaining representative graph characteristics, these sensitivity analyses were conducted on subgraphs sampled at 10% of the original graph size for the Cora, Yelp, and PubMed datasets.

**Impact of Pruning Ratio.** Table 14 presents the detailed accuracy and latency figures as the pruning ratio (percentage of nodes removed) is varied from 10% to 90%, while keeping the polynomial degrees for adaptive activations fixed (e.g., using PSet2: (5,3,2) representing medium fidelity).

As observed in the main paper and further detailed in Table 14, increasing the pruning ratio consistently leads to a reduction in inference latency across all three dataset subsets. This is an expected outcome, as processing fewer nodes and edges directly translates to fewer homomorphic operations. For instance, on the Cora subset, increasing the pruning ratio from 10% to 50% reduces latency from 93.47s to 64.94s. Concurrently, accuracy generally declines with increased pruning, as more graph information is discarded. However, the rate of decline varies. For Cora and PubMed subsets, a pruning ratio up to 40-50% results in a relatively graceful accuracy degradation, suggesting that a significant portion of nodes can be removed while retaining substantial predictive power. Yelp

Table 14: Impact of Pruning Ratio on DESIGN Performance. Acc. is Accuracy (%), Lat. is Latency (s).

| Pruning | Cora | | Yelp | | PubMed | |
|---|---|---|---|---|---|---|
| Ratio (%) | Acc. | Lat. | Acc. | Lat. | Acc. | Lat. |
| 10 | 76.65 | 93.47 | 53.05 | 17.00 | 56.92 | 27.89 |
| 20 | 75.96 | 87.08 | 52.89 | 15.53 | 56.40 | 25.44 |
| 30 | 75.50 | 79.13 | 52.00 | 14.38 | 56.05 | 23.50 |
| 40 | 74.19 | 70.01 | 51.61 | 12.92 | 55.02 | 21.42 |
| 50 | 72.03 | 64.94 | 50.06 | 11.88 | 53.90 | 19.24 |
| 60 | 70.47 | 58.86 | 48.28 | 10.43 | 51.83 | 17.19 |
| 70 | 65.68 | 48.90 | 45.41 | 9.31 | 48.72 | 15.20 |
| 80 | 57.55 | 44.15 | 39.92 | 7.89 | 42.77 | 13.11 |
| 90 | 47.06 | 37.36 | 32.92 | 6.71 | 35.30 | 10.89 |

appears more sensitive, with accuracy dropping more sharply at higher pruning ratios. This analysis underscores DESIGN's capability to achieve substantial speedups (e.g., more than halving latency at 70-80% pruning on Cora) and highlights the importance of selecting an appropriate pruning ratio based on the specific dataset's tolerance and the desired accuracy-latency trade-off. The results demonstrate that DESIGN effectively identifies and removes redundancy, allowing for a tunable balance between efficiency and utility.

**Impact of Polynomial Degrees for Adaptive Activation.** For the sensitivity to polynomial activation degrees, the main paper (Section 4.5, Table 3) already presents the core findings using a fixed pruning ratio (e.g., 40%) and three polynomial degree sets: PSet1 (High-fidelity: (7,5,3)), PSet2 (Medium-fidelity: (5,3,2)), and PSet3 (Low-fidelity: (3,2,1)). The discussion in the main paper highlights that higher-degree sets (PSet1) yield better accuracy due to superior approximation of non-linearities but incur higher latency due to increased FHE multiplication costs. Conversely, lower-degree sets (PSet3) reduce latency at the cost of some accuracy. The strength of DESIGN lies in its adaptive allocation mechanism. By assigning these varied-degree polynomials based on pre-determined node importance (derived from the FHE-compatible degree scoring and partitioning stage), our framework effectively balances the need for high accuracy on critical nodes (utilizing higher-degree polynomials like those in PSet1 for the most important level) with the imperative for efficiency on less important nodes (utilizing lower-degree polynomials like those in PSet3 for the least important retained level). This adaptive strategy allows DESIGN to optimize the overall cost-accuracy profile more effectively than a uniform polynomial degree approach, showcasing its ability to tailor computational complexity to information value. For example, using PSet2 as a balanced set for adaptive allocation typically provides a good compromise, achieving accuracy close to PSet1 while offering latency significantly better than PSet1 and closer to PSet3, demonstrating the practical benefit of the adaptive scheme.

**Sensitivity to importance metric under encryption.** We compare degree with a feature L2-norm score on a Cora variant that includes synthetic bridge nodes. As reported in Table 15, the L2-norm achieves slightly higher accuracy (+0.6 pp) but increases end-to-end latency from 0.71 s to 3.97 s and raises the net multiplicative depth from 10 to 12 because it requires ciphertext–ciphertext multiplications and extra rotations. This trade-off indicates that, under current CKKS costs, degree remains the most practical metric because it preserves most of the accuracy benefits while keeping the depth and latency low.

Table 15: Sensitivity to importance metric on a "Bridge-Nodes" Cora variant (40% pruning, same CKKS params unless depth forces longer chain).

| Metric | Accuracy (%) | End-to-end latency (s) | Net depth | Extra HE ops vs. degree |
|---|---|---|---|---|
| Node degree | 82.4 | 0.71 | 10 | — |
| Feature L2-norm | 83.0 | 3.97 | 12 | +1 Rot & +1 C×C Mult / node |

---

**Algorithm 1** FHE-Compatible Importance Mask Generation (CKKS Focus)

---

**Require:** Encrypted adjacency matrix $\tilde{\mathbf{A}}$ under CKKS; Importance thresholds $\boldsymbol{\tau} = [\tau_1, \ldots, \tau_m]$ (plaintext or encrypted, with $\tau_0 = \infty$ and $\tau_1 > \cdots > \tau_m$).
**Ensure:** Encrypted prune mask $\tilde{M}_0$ and level masks $\tilde{M}_1, \ldots, \tilde{M}_m$.
1: Compute encrypted node degree score vector $\tilde{\mathbf{s}}$ from $\tilde{\mathbf{A}}$ using HE.Add and potentially HE.Rotate.
2: Initialize encrypted mask vectors: $\tilde{M}_0, \tilde{M}_1, \ldots, \tilde{M}_m \leftarrow \text{Enc}(\mathbf{0})$ (vectors of size $n$).
3: $\tilde{\mathbf{1}} \leftarrow \text{Enc}(\mathbf{1})$ (vector of encrypted ones of size $n$).
   // Generate prune mask for nodes with score below $\tau_m$
4: $\tilde{\tau}_m^{\text{enc}} \leftarrow \text{EnsureEncrypted}(\tau_m)$            $\triangleright$ Ensure threshold is encrypted if not already
5: $\tilde{M}_0 \leftarrow \tilde{\mathbf{1}} \ominus \text{HE.AprxCmp}(\tilde{\mathbf{s}}, \tilde{\tau}_m^{\text{enc}})$            $\triangleright$ $\tilde{M}_{0,v} \approx \text{Enc}(1)$ if $s_v < \tau_m$
   // Generate level masks for retained nodes
6: **for** $i \leftarrow 1$ to $m$ **do**
7:      $\tilde{\tau}_i^{\text{enc}} \leftarrow \text{EnsureEncrypted}(\tau_i)$
8:      $\tilde{\tau}_{i-1}^{\text{enc}} \leftarrow \text{EnsureEncrypted}(\tau_{i-1})$      $\triangleright$ Assuming $\tau_0 = \infty$ handled appropriately by HE.AprxCmp
9:      ge_tau_i $\leftarrow$ HE.AprxCmp($\tilde{\mathbf{s}}, \tilde{\tau}_i^{\text{enc}}$)            $\triangleright \approx \text{Enc}(1)$ if $s_v \geq \tau_i$
10:     lt_tau_i_minus_1 $\leftarrow \tilde{\mathbf{1}} \ominus$ HE.AprxCmp($\tilde{\mathbf{s}}, \tilde{\tau}_{i-1}^{\text{enc}}$)     $\triangleright \approx \text{Enc}(1)$ if $s_v < \tau_{i-1}$
11:     $\tilde{M}_i \leftarrow$ ge_tau_i $\odot$ lt_tau_i_minus_1      $\triangleright$ $\tilde{M}_{i,v} \approx \text{Enc}(1)$ if $\tau_i \leq s_v < \tau_{i-1}$
12: **end for**
13: **return** $\tilde{M}_0, \tilde{M}_1, \ldots, \tilde{M}_m$

---

## C   DETAILED EXPERIMENTAL SETTINGS

### C.1   ALGORITHMIC ROUTINE AND DISCUSSION

This section provides the detailed algorithmic routine for the two core algorithmic stages of our proposed DESIGN framework: (1) FHE-compatible importance mask generation (Algorithm 1), and (2) the subsequent FHE GNN inference that incorporates these masks for graph pruning and adaptive activation allocation (Algorithm 2). Both algorithms are designed to operate entirely on the server side, processing client-encrypted graph data under the CKKS FHE scheme to preserve data privacy throughout the inference pipeline.

Algorithm 1 outlines the server-side procedure for dynamically generating encrypted importance masks based on the input encrypted graph. The process begins with computing encrypted node degree scores from the encrypted adjacency matrix $\tilde{\mathbf{A}}$, a metric chosen for its FHE efficiency as it primarily relies on homomorphic additions. These encrypted scores $\tilde{\mathbf{s}}$ are then compared against a set of $m$ pre-defined importance thresholds $\boldsymbol{\tau}$. Since direct comparisons are not native to CKKS, this step utilizes an approximate homomorphic comparison operator (HE.AprxCmp), which typically evaluates a polynomial approximation of a comparison function. This process yields an encrypted prune mask $\tilde{M}_0$, identifying nodes deemed unimportant (score below $\tau_m$), and $m$ encrypted level masks $\tilde{M}_1, \ldots, \tilde{M}_m$, categorizing the retained nodes into distinct importance levels. While the HE.AprxCmp operations are computationally intensive due to their multiplicative depth, this stage is crucial for enabling the subsequent adaptive optimizations. The function 'EnsureEncrypted' is a conceptual step indicating that plaintext thresholds are appropriately encoded and encrypted if the comparison is between two ciphertexts, or that plaintext thresholds are directly used if the FHE library supports ciphertext-plaintext comparison via HE.AprxCmp.

Algorithm 2 details the FHE GNN inference procedure, which leverages the masks generated by Algorithm 1. Initially, the prune mask $\tilde{M}_0$ is applied to the input encrypted graph $\tilde{\mathcal{G}} = (\tilde{\mathbf{A}}, \tilde{\mathbf{X}})$ to logically remove nodes and edges. This is achieved by homomorphically multiplying the features $\tilde{\mathbf{X}}$ and adjacency entries $\tilde{\mathbf{A}}$ with a "keep mask" derived from $\tilde{M}_0$, effectively zeroing out the contributions of pruned elements and resulting in a pruned graph representation $(\tilde{\mathbf{A}}', \tilde{\mathbf{X}}')$. The GNN inference then proceeds layer by layer using this pruned structure. For each layer, after computing the encrypted pre-activations $\tilde{\mathbf{Z}}^{(l+1)}$ (as per Equation 4 in the main text, involving homomorphic matrix operations), the adaptive activation mechanism is invoked. The encrypted level masks $\tilde{M}_1, \ldots, \tilde{M}_m$ are used to

---

**Algorithm 2** FHE GNN Inference with Pruning and Adaptive Activation (CKKS Focus)

---

**Require:** Encrypted Graph $\tilde{\mathcal{G}} = (\tilde{\mathbf{A}}, \tilde{\mathbf{X}})$ under CKKS; Pre-trained GNN model $f_{\boldsymbol{\theta}}$ with $L$ lay-
ers; Encrypted masks $\tilde{M}_0, \tilde{M}_1, \ldots, \tilde{M}_m$ from Algorithm 1; Polynomial activation functions
$\{P_{d_i}(z)\}_{i=1}^m$ with degrees $\mathbf{d} = [d_1, \ldots, d_m]$ $(d_1 > \cdots > d_m)$.

**Ensure:** Encrypted inference result $\tilde{\mathbf{Y}}$.
     // Apply prune mask $\tilde{M}_0$ to input features and adjacency matrix
 1:  $\tilde{\mathbf{1}} \leftarrow \text{Enc}(\mathbf{1})$ (vector of encrypted ones of size $n$)
 2:  $\text{Keep}\tilde{\text{Mask}} \leftarrow \tilde{\mathbf{1}} \ominus \tilde{M}_0$          $\triangleright$ Mask for nodes to keep, $\text{Keep}\tilde{\text{Mask}}_v \approx \text{Enc}(1)$ if $s_v \geq \tau_m$
 3:  $\tilde{\mathbf{X}}' \leftarrow \tilde{\mathbf{X}} \odot \text{Keep}\tilde{\text{Mask}}$                $\triangleright$ Zero out features of pruned nodes
 4:  // For each entry $(u, v)$ in $\tilde{\mathbf{A}}$:
 5:  $\tilde{A}'_{uv} \leftarrow \tilde{A}_{uv} \odot \text{Keep}\tilde{\text{Mask}}_u \odot \text{Keep}\tilde{\text{Mask}}_v$     $\triangleright$ Remove edges connected to pruned nodes
 6:  Let $\tilde{\mathbf{A}}'$ be the resulting pruned encrypted adjacency matrix.
     // Initialize hidden states
 7:  $\tilde{\mathbf{H}}^{(0)} \leftarrow \tilde{\mathbf{X}}'$
     // Perform GNN layer computations
 8:  **for** layer $l \leftarrow 0$ to $L - 1$ **do**
 9:     // Compute encrypted pre-activations using the pruned graph structure
10:     $\tilde{\mathbf{Z}}^{(l+1)} \leftarrow \left( \tilde{\mathbf{A}}' \otimes \tilde{\mathbf{H}}^{(l)} \otimes \mathbf{W}_1^{(l)} \right) \oplus \left( \tilde{\mathbf{H}}^{(l)} \otimes \mathbf{W}_2^{(l)} \right) \oplus \mathbf{b}^{(l)}$ (as per Eq. 4)
11:     // Apply adaptive polynomial activations homomorphically
12:     $\tilde{\mathbf{H}}^{(l+1)} \leftarrow \text{Enc}(\mathbf{0})$              $\triangleright$ Initialize output activations for layer $l + 1$
13:     **for** $i \leftarrow 1$ to $m$ **do**
14:         $\tilde{\mathbf{P}}_{d_i}(\tilde{\mathbf{Z}}^{(l+1)}) \leftarrow \text{HE.PolyEval}(P_{d_i}, \tilde{\mathbf{Z}}^{(l+1)})$  $\triangleright$ Evaluate polynomial $P_{d_i}$ for all nodes
15:         $\tilde{\mathbf{H}}^{(l+1)} \leftarrow \tilde{\mathbf{H}}^{(l+1)} \oplus (\tilde{M}_i \odot \tilde{\mathbf{P}}_{d_i}(\tilde{\mathbf{Z}}^{(l+1)}))$    $\triangleright$ Selectively add based on level mask $\tilde{M}_i$
16:     **end for**
17:  **end for**
18:  **return** $\tilde{\mathbf{Y}} = \tilde{\mathbf{H}}^{(L)}$

---

selectively apply different pre-defined polynomial activation functions $P_{d_i}$ (with varying degrees $d_i$) to the pre-activations of nodes based on their assigned importance level. This selective application is achieved through homomorphic multiplications with the masks and summation of the masked polynomial evaluation results, ensuring that more critical nodes receive higher-fidelity (higher-degree) activations while less critical nodes use computationally cheaper (lower-degree) ones. The final output is the encrypted GNN inference result $\tilde{\mathbf{Y}}$.

## C.2   FHE Scheme and Cryptographic Parameters

For all homomorphic encryption operations within our DESIGN framework and the FHE baselines, we employ the Cheon-Kim-Kim-Song (CKKS) scheme (Cheon et al., 2017), renowned for its efficacy in handling approximate arithmetic on real-valued data, a common requirement in machine learning applications. The specific cryptographic parameters are chosen to ensure a security level of at least 128 bits, adhering to established cryptographic standards (Albrecht et al., 2021), while simultaneously supporting the necessary multiplicative depth for our GNN inference pipeline, including the approximate comparison and polynomial activation evaluation stages. We set the polynomial modulus degree $N$ to $2^{15}$ (or $2^{16}$ for experiments requiring greater depth or precision, specified per experiment if varied). The coefficient moduli $q_i$ in the RNS (Residue Number System) representation are carefully selected to accommodate the noise growth throughout the computation. Specifically, we use a chain of prime moduli, including special primes for key-switching and rescaling operations, with bit-lengths typically around 50-60 bits for data moduli and a larger special prime. The initial scale factor for encoding plaintext values into polynomials is set to $2^{40}$ (or adjusted as needed, e.g., $2^{30}$ as mentioned in implementation details, to balance precision and noise). Rescaling operations are performed after homomorphic multiplications to manage the scale of ciphertexts and control noise propagation, typically reducing the scale by one prime modulus. The choice of these parameters, particularly the total bit-length of the coefficient modulus chain, directly determines the maximum multiplicative depth supported by the leveled FHE computation before bootstrapping would be required. Our experiments are designed to operate within this leveled FHE paradigm to avoid the substantial overhead associated

with bootstrapping in current CKKS implementations. All FHE operations are performed using the Microsoft SEAL library (Max).

### C.3 GNN MODEL ARCHITECTURES AND TRAINING

For all experiments, we employed a consistent Graph Neural Network (GNN) architecture designed for simplicity and compatibility with FHE operations. This model consists of two Graph Convolutional Layers (GCNConv). The first GCNConv layer takes the input node features, with dimensionality specific to each dataset, and transforms them into a hidden representation of dimensionality 2. Following this layer, we apply a quadratic activation function. This choice is motivated by its exact polynomial form, which avoids the need for further approximation under FHE, unlike common activations such as ReLU or GELU, thereby simplifying the homomorphic evaluation and reducing potential approximation errors. A dropout layer with a rate of 0.5 is then applied to the activated hidden representations to mitigate overfitting. The second GCNConv layer takes these 2-dimensional representations as input and maps them to an output dimensionality corresponding to the number of target classes for the specific node classification task of each dataset.

This two-layer GCN architecture was uniformly applied across all five benchmark datasets to ensure a fair comparison of our framework's performance. Crucially, all GNN models were pre-trained in a standard plaintext (unencrypted) environment on their respective training splits before being utilized for the encrypted inference experiments. This pre-training strategy is standard in FHE inference literature, as it circumvents the extreme computational expense and complexity associated with training GNNs directly on encrypted data, while still allowing for the evaluation of privacy-preserving inference on models with learned weights.

The plaintext training was conducted using the Adam optimizer with a learning rate set to 0.01 and a weight decay (L2 regularization) of $5 \times 10^{-4}$ to prevent overfitting. Models were trained for a maximum of 200 epochs, with an early stopping mechanism based on performance on a separate validation set to select the best performing model parameters. The aforementioned dropout rate of 0.5 applied after the first layer's activation function also contributed to model generalization during this pre-training phase. For the loss function during pre-training, we utilized cross-entropy loss for all single-label node classification tasks (Cora, citepseer, PubMed). For the datasets involving multi-label classification, specifically ogbn-proteins and Yelp, we employed binary cross-entropy with logits loss (BCEWithLogitsLoss), which is appropriate for scenarios where each node can belong to multiple classes simultaneously. The choice of evaluation metrics for reporting performance was also dataset-dependent: standard classification accuracy was used for the single-label benchmarks, while the Area Under the Receiver Operating Characteristic Curve (ROC AUC) was employed for the multi-label datasets, providing a robust measure of performance in such settings.

### C.4 IMPLEMENTATION OF FRAMEWORK COMPONENTS

Our DESIGN framework's components are implemented for the CKKS FHE scheme, with specific operational details tailored for reproducibility. For FHE-compatible importance scoring, encrypted node degrees $\tilde{\mathbf{s}}$ are computed from the encrypted adjacency matrix $\tilde{\mathbf{A}}$. This involves representing $\tilde{\mathbf{A}}$ using its encrypted diagonals and performing a homomorphic aggregation with an encrypted vector of ones ($\tilde{\mathbf{1}}$) using our '_aggregate' function, which primarily leverages `HE.Add`, `HE.Mult`, and `HE.Rotate` for efficient sum-product operations. Homomorphic partitioning and mask generation then compare these encrypted scores $\tilde{\mathbf{s}}$ against pre-defined importance thresholds $\boldsymbol{\tau}$ (e.g., '[5.0, 2.0]' for two retained levels plus a prune level). This comparison is facilitated by our `HE.AprxCmp` operator, implemented as the homomorphic evaluation ('_eval_poly') of a fixed comparison polynomial, $P_{\mathrm{cmp}}(x) = -0.25x^3 + 0.75x + 0.5$, on the encrypted difference between scores and thresholds. This process, detailed in Algorithm 1, yields the encrypted prune mask $\tilde{M}_0$ and level masks $\tilde{M}_1, \ldots, \tilde{M}_m$. Subsequently, homomorphic graph pruning applies the $\tilde{M}_0$ mask by deriving a keep mask ($\tilde{\mathbf{1}} \ominus \tilde{M}_0$) and performing homomorphic element-wise multiplication ($\odot$) with the initial encrypted node features $\tilde{\mathbf{X}}$ and each encrypted diagonal of $\tilde{\mathbf{A}}$, as outlined in Algorithm 2 (Lines 3-6). Finally, the adaptive polynomial activation mechanism employs pre-defined polynomial functions $\{P_{d_i}\}$, whose plaintext coefficients are specified (e.g., '[[0,0,1.0]]' for $P_2(x) = x^2$, '[0,1.0]' for $P_1(x) = x$). The homomorphic evaluation of these polynomials (`HE.PolyEval`) on encrypted pre-activations $\tilde{\mathbf{Z}}^{(l+1)}$ is performed using an efficient Horner's method implementation in our '_eval_poly' function. The

final activated output $\tilde{\mathbf{H}}^{(l+1)}$ is a masked sum using the level masks $\tilde{M}_i$, as detailed in Algorithm 2 (Lines 12-16). The specific thresholds $\tau$ and polynomial coefficient sets 'poly_configs_D_coeffs' used for each experimental setting are provided alongside their respective results.

## C.5 EXPERIMENTAL ENVIRONMENT AND REPRODUCIBILITY

All empirical evaluations were performed on a dedicated server infrastructure to ensure consistent and reproducible results. The server is equipped with an AMD EPYC 7763 64-Core Processor and 1007 GB of system memory (RAM). While our FHE computations are primarily CPU-bound, the system also includes two NVIDIA RTX 6000 Ada Generation GPUs and one NVIDIA A100 80GB PCIe GPU, which were utilized for plaintext model training and baseline evaluations where applicable.

The software environment was standardized across all experiments. Key libraries and their versions include Python 3.10.17, PyTorch 2.4.0 for neural network operations, and PyTorch Geometric (PyG) 2.6.1 for graph-specific functionalities. For Fully Homomorphic Encryption, our implementations and baselines utilized OpenFHE version 1.2.3 and Microsoft SEAL version 4.0.0, depending on the specific FHE scheme or baseline being evaluated.

To account for inherent variability, each reported experimental result represents the mean value obtained from 5 independent runs. Reproducibility was further promoted by employing a fixed random seed of 42 for all stochastic processes, including data splitting (where applicable) and model weight initialization during pre-training.

## C.6 FHE COMPLEXITY OF STATISTICAL METRICS

The choice of statistical metric for assessing node importance under FHE is heavily constrained by computational feasibility. Table 16 provides a qualitative comparison of the resources required to compute common graph statistics directly on encrypted data using the CKKS scheme. This comparison considers the primary homomorphic operations involved, the minimum multiplicative depth incurred (a critical factor for noise management and parameter selection in leveled FHE), and the overall relative complexity, which encompasses both computational time and noise growth.

Table 16: Qualitative Comparison of Computing Graph Statistics under CKKS FHE. Assumes CKKS scheme. Multiplicative Depth (Mult. Depth) indicates minimum required multiplicative levels. Complexity reflects relative computational cost and noise growth.

| Statistical Metric | Required HE Operations | Mult. Depth | Complexity |
|---|---|---|---|
| Node Degree ($\sum A_{vu}$) | `HE.Add`, `HE.Rotate` | Low (0-1) | Low |
| Feature Mean ($\text{mean}(\mathbf{x}_v)$) | `HE.Add`, `HE.Mult` (ptxt), `HE.Rotate` | Low (0-1) | Low-Medium |
| Feature L2 Norm$^2$ ($\|\mathbf{x}_v\|_2^2$) | `HE.Mult`, `HE.Add`, `HE.Rotate` | Medium ($\geq 1$) | High |
| Feature Variance | `HE.Mult`, `HE.Add`/$\oplus$, `HE.Mult` (ptxt), `HE.Rotate` | High ($\geq 2$) | Very High |

The analysis presented in Table 16 clearly illustrates that metrics requiring homomorphic multiplication (`HE.Mult`), such as the squared L2 norm of node features or feature variance, are substantially more demanding. These operations not only increase the direct computational time but also contribute significantly to noise growth in ciphertexts and necessitate a greater multiplicative depth. Managing this increased depth often requires larger FHE parameters, which further slows down all homomorphic operations. In contrast, node degree, computed primarily through homomorphic additions (`HE.Add`) and potentially rotations (`HE.Rotate` for SIMD processing), exhibits low multiplicative depth and overall complexity. Similarly, computing the feature mean involves additions and efficient plaintext multiplications. This stark difference in FHE computational cost is the primary motivation for our framework's adoption of node degree as the statistical indicator for importance scoring in the initial FHE-compatible stage, as discussed in Section 3.1. While richer feature-based statistics might offer more nuanced importance measures in plaintext, their prohibitive FHE overhead could easily outweigh any benefits derived from more precise pruning if computed directly on ciphertexts in an online fashion. Our design prioritizes a lightweight initial scoring mechanism to ensure the overall pruning and adaptive inference pipeline remains efficient.

