# OpenReview forum: "DESIGN: Encrypted GNN Inference via Server-Side Input Graph Pruning"
_ICLR.cc/2026/Conference — Submitted to ICLR 2026_

### Official Review · Reviewer_YBd8 · 2025-10-31

**Soundness:** 2
**Presentation:** 2
**Contribution:** 2
**Rating:** 4
**Confidence:** 4

**Summary:**

This paper proposes DESIGN, a server-side method for homomorphically encrypted GNN inference. The key idea is to estimate node importance directly over encrypted graphs via degree statistics and approximate homomorphic comparisons, enabling pruning of low-importance structure. It also assigns polynomial activation degrees based on importance levels to reduce costly multiplications. Experiments over multiple benchmarks report noticeable latency reductions while maintaining accuracy in a comparable range to recent FHE-GNN systems.

**Strengths:**

1. This paper addresses a meaningful problem setting: fully server-side FHE-GNN inference without client modifications.
2. Experimental results show that the proposed method can achieve consistent latency improvements across multiple datasets.
3. This paper includes ablations and sensitivity analysis to illustrate cost-accuracy trade-offs.
4. The proposed framework is modular and could integrate with various GNN layers.

**Weaknesses:**

1. The “degree” to “importance” assumption is weakly motivated and likely task-misaligned. Accuracy drops indicate that important information is pruned away.
2. The degradation pattern suggests structural brittleness that without a more expressive encrypted importance estimator, the system cannot improve through incremental refinement.
3. Contribution remains primarily system-level optimization, without a clear conceptual advance.

**Questions:**

1.	What empirical evidence supports degree as a reliable proxy for task importance under encrypted inference? Have you compared against any other encrypted-feasible statistics (e.g., weighted degree, simple encrypted feature norms)?
2.	How does the method behave on graphs where degree is anti-correlated with classification relevance (e.g., heterophilic graphs)?
3.	In Table 2: why does SEAL achieve 68.10% on Cora while OpenFHE drops to 28.00%, despite both being foundational FHE implementations?

---

### Official Review · Reviewer_A5uV · 2025-10-31

**Soundness:** 2
**Presentation:** 3
**Contribution:** 3
**Rating:** 4
**Confidence:** 3

**Summary:**

This paper addresses the significant computational overhead of performing Graph Neural Network (GNN) inference on sensitive data under Fully Homomorphic Encryption (FHE). The high cost of homomorphic operations makes direct application of FHE prohibitively slow for practical use. The authors propose DESIGN, a server-side framework that accelerates encrypted GNN inference without requiring any client-side modifications.
The core contribution is a novel "dual-pruning" scheme that operates entirely on encrypted data. First, the framework computes an FHE-compatible importance score for each node (using encrypted node degree as an efficient proxy). Second, it uses approximate homomorphic comparison protocols to partition nodes into multiple importance levels, generating a set of encrypted masks. These masks then drive two concurrent optimizations during inference: (1) the logical pruning of unimportant nodes and their incident edges, reducing the volume of data to be processed, and (2) an adaptive activation mechanism that assigns computationally cheaper, low-degree polynomial activation functions to less important nodes while reserving more accurate, high-degree polynomials for critical ones. The authors claim that across several benchmark datasets, DESIGN substantially accelerates inference (e.g., 1.67x-2.39x speedup over a SEAL baseline) while maintaining competitive accuracy.

**Strengths:**

1.	The paper tackles an important and timely problem—improving the practicality of encrypted GNN inference under FHE—and proposes a coherent, end-to-end framework that operates entirely on the server side.
2.	The dual-pruning design is technically elegant and well-justified: a single encrypted importance metric (degree) drives both pruning and adaptive activation allocation, which efficiently balances latency and multiplicative depth.

**Weaknesses:**

1.	The pruning masks are generated dynamically at runtime for each encrypted graph, meaning computation patterns (e.g., number of masked nodes or chosen activation degrees) depend on input data. This introduces potential side-channel leakage through timing or resource-usage patterns, which could reveal structural information such as degree distribution. The paper does not acknowledge or mitigate this risk, leaving a gap in its privacy analysis.
2.	The choice of node degree as the sole importance metric is efficient but simplistic. The authors justify it by low homomorphic cost, yet DESIGN has not been tested on graphs where degree poorly correlates with node importance (e.g., heterophilic or feature-dominant graphs). The generality of the approach is therefore unverified.
3.	The trade-off between the initial HE.AprxCmp partitioning cost and subsequent latency savings is not quantitatively analyzed. While Table 3 includes partial timing information, there is no full phase-by-phase breakdown or multiplicative-depth accounting to confirm where gains originate or when the upfront cost is amortized.
4.	The “Pruning-Only” variant sometimes achieves higher accuracy than the baseline, but the explanation remains speculative. No diagnostic analysis (e.g., degree quantiles or label consistency of pruned nodes) is provided to substantiate the claim that pruning removes noisy neighborhoods.
5.	Scalability remains unclear. All evaluations use small graphs (≤10K nodes) and shallow architectures (two-layer GCNs). Although the authors acknowledge that HE.AprxCmp has quadratic complexity in node count, they provide no empirical or theoretical scaling analysis to indicate practical limits.
6.	The evaluation section contains predictive wording (“expected,” “anticipated”) in places where results have already been measured (e.g., Table 1). This inconsistent tone weakens the paper’s empirical rigor.
7.	The notion of “importance” is narrowly defined. The authors do not discuss how importance could be generalized to incorporate features or learned weights while maintaining FHE efficiency.

**Questions:**

1.	Have the authors considered possible side-channel leakage due to data-dependent computation patterns (runtime pruning and adaptive activation)?
Could a server observing timing or resource usage infer graph structural information (e.g., degree distribution)?
2.	Could you provide a clearer runtime and multiplicative-depth breakdown across the major stages—HE.AprxCmp and masking versus per-layer aggregation and activation?
This would help verify that the initial cost is offset by the subsequent gains.
3.	How sensitive is DESIGN to the node-degree assumption?
Have you tested or can you discuss cases (e.g., heterophilic graphs, feature-dominant tasks) where degree may not correlate with importance?
4.	The “Pruning-Only” variant improves accuracy over the baseline on some datasets.
Can you analyze which nodes are pruned (e.g., degree quantile, label homophily) to validate whether this acts as regularization?
5.	Were iteration limits and encryption parameters consistent across all baselines (SEAL, Penguin, LinGCN, CryptoGCN)?
If not, please clarify the resource budgets to justify fairness.
6.	What are the practical scalability limits of the homomorphic comparison stage?
At what graph size or number of partitions does its cost dominate overall latency?
7.	How would DESIGN generalize to deeper GNNs or different architectures (GraphSAGE, GAT) that use multi-hop or attention-based aggregation?
8.	Could you discuss how “importance” might be extended or learned under FHE while maintaining efficiency, for instance by mixing structural and feature-based metrics?
9. Others see Weakness.

---

### Official Review · Reviewer_8TpL · 2025-10-31

**Soundness:** 2
**Presentation:** 3
**Contribution:** 1
**Rating:** 2
**Confidence:** 4

**Summary:**

The paper DESIGN proposes an efficient framework for encrypted GNN inference under fully homomorphic encryption. It introduces server-side input graph pruning and adaptive polynomial activations to reduce the computational cost of FHE-based operations. Node degree is used as an encrypted importance indicator, enabling structure-aware pruning and selective activation without decrypting the graph.

**Strengths:**

1. tackles the practical challenge of efficient GNN inference under fully homomorphic encryption (FHE).
2.Clearly defines the computation and accuracy trade-off in encrypted GNNs and provides quantitative evidence.

**Weaknesses:**

(1)Lack of justification for using node degree as the importance metric.
 The use of node degree as the only importance metric is heuristic and may not reflect true semantic importance.
(2)Missing analysis on sensitivity or correlation.
Although Appendix C analyzes pruning thresholds, there is no quantitative analysis showing how well degree-based importance aligns with actual contribution to model accuracy or message propagation strength.
(3) The adaptive activation scheme lacks theoretical justification and error analysis.
(4) The encrypted degree computation may still leak structural patterns statistically, posing privacy risks.

**Questions:**

See weakness above.

---

> ### Comment · Reviewer_8TpL · 2025-11-26
>
> Since there is no rebuttal or response, I decide to maintain my current score.

---

### Official Review · Reviewer_scBn · 2025-10-31

**Soundness:** 2
**Presentation:** 3
**Contribution:** 2
**Rating:** 2
**Confidence:** 5

**Summary:**

DESIGN presents an efficient framework for performing GNN inference on fully homomorphically encrypted data, enabling privacy-preserving computation through server-side graph pruning and adaptive processing. The framework securely estimates node importance scores under encryption to prune less significant regions of the graph, thereby reducing redundant homomorphic operations. Additionally, it enhances efficiency by dynamically selecting polynomial activation functions of different degrees according to each node’s importance.

**Strengths:**

The research problem is both meaningful and timely, as reducing inference latency remains a critical challenge in privacy-preserving GNN computation. The proposed approach of offloading computation to the server side is practical and well-aligned with real-world deployment scenarios, making the framework readily adaptable for practical implementation.

**Weaknesses:**

In the evaluation section, the latency breakdown is insufficiently detailed, it remains unclear how much of the total runtime is attributed to graph pruning versus HE computation within the GNN. Furthermore, the process of generating encrypted graph masks appears to introduce additional HE operation complexity compared to prior methods. This aspect should be further clarified and quantitatively analyzed by the authors to better understand its impact on overall performance and scalability.

With the proposed method, the pruned graph produces encrypted binary masks. Although these masks represent Boolean outcomes based on thresholding, it is unclear how the encrypted zeros can practically contribute to accelerating inference. Further clarification is needed on how computation on these encrypted masks leads to latency reduction?

**Questions:**

see weakness.

---

### Meta-Review · Area_Chair_9RNv · 2026-01-04

**Summary:**

This paper proposes DESIGN, a server-side framework combining encrypted degree-based pruning with importance-aware adaptive polynomial activations. Reviewers generally agree that the idea of driving both pruning and activation allocation from a single encrypted importance signal is practically appealing, and the system-level integration is well executed.

However, the reviews also surface several substantial concerns that limit the paper’s impact and readiness. A recurring issue is the reliance on node degree as the sole importance metric: this choice is seen as heuristic, weakly justified, and potentially misaligned with task relevance, especially on heterophilic or feature-dominant graphs. The paper lacks quantitative evidence correlating degree-based importance with actual contribution to accuracy or message passing, and alternative encrypted-feasible metrics are not explored. Relatedly, the adaptive activation scheme lacks theoretical grounding or error analysis.

From a systems and privacy perspective, reviewers note insufficient runtime decomposition and scalability analysis. It remains unclear how the cost of encrypted comparisons and mask generation amortizes against downstream savings, and evaluations are limited to relatively small graphs and shallow GNNs. Multiple reviewers also raise concerns about potential side-channel leakage due to data-dependent pruning and activation patterns, which is not addressed in the privacy analysis.

The authors chose to not submit a rebuttal. Given the outstanding concerns, I recommend rejection.

**Reviewer Concerns:**

The authors chose not to submit a rebuttal.

**Reviewer Scores:**

The authors chose not to submit a rebuttal. Hence, the original scores stand, all of which advoate rejection.

---

### Decision · Program_Chairs · 2026-01-26

Reject